# "*Don't Take This Out of Context!*"
# On the Need for Contextual Models and Evaluations for Stylistic Rewriting

**Akhila Yerukola**[♡]    **Xuhui Zhou**[♡]    **Elizabeth Clark**[◇]    **Maarten Sap**[♡♣]

[♡]Language Technologies Institute, Carnegie Mellon University
[◇]Google DeepMind  [♣]Allen Institute for AI

✉ ayerukol@andrew.cmu.edu

## Abstract

Most existing stylistic text rewriting methods and evaluation metrics operate on a sentence level, but ignoring the broader context of the text can lead to preferring generic, ambiguous, and incoherent rewrites. In this paper, we investigate integrating the preceding textual context into both the *rewriting* and *evaluation* stages of stylistic text rewriting, and introduce a new composite contextual evaluation metric `CtxSimFit` that combines similarity to the original sentence with contextual cohesiveness. We comparatively evaluate non-contextual and contextual rewrites in formality, toxicity, and sentiment transfer tasks. Our experiments show that humans significantly prefer contextual rewrites as more fitting and natural over non-contextual ones, yet existing sentence-level automatic metrics (e.g., ROUGE, SBERT) correlate poorly with human preferences ($\rho$=0–0.3). In contrast, human preferences are much better reflected by both our novel `CtxSimFit` ($\rho$=0.7–0.9) as well as proposed context-infused versions of common metrics ($\rho$=0.4–0.7). Overall, our findings highlight the importance of integrating context into the generation and especially the evaluation stages of stylistic text rewriting.

## 1 Introduction

Existing methods for *stylistic text rewriting*, i.e., adapting the text to a particular style while preserving its originally intended meaning, often fail to account for a statement's context (e.g., Hu et al., 2017; Shen et al., 2017; Fu et al., 2018; Li et al., 2018; Lample et al., 2019; Madaan et al., 2020; Hallinan et al., 2023). As a result, these systems may change the speakers' original communicative intents and generate contextually irrelevant and generic outputs. For example, in Figure 1, a non-contextual model rewriting an informal response to a formal one simply replaces words with more formal synonyms, whereas a contextual rewriting model can use the broader conversational context to produce a more specific and natural formal rewrite.

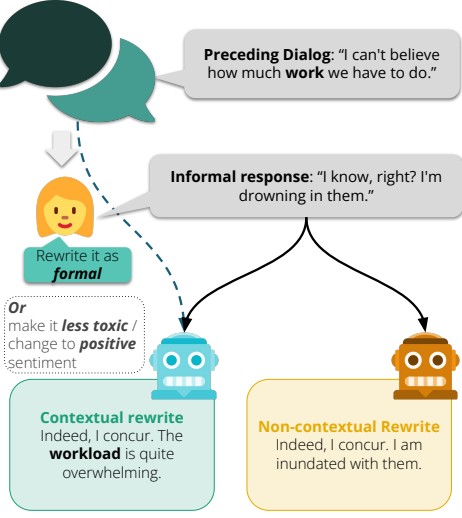

Figure 1: Example of using the preceding dialog utterance to help with stylistic rewriting: here, we transform an informal response into formal language. Incorporating "workload" and "overwhelming" enhances the contextual cohesiveness of the rewritten text, while solely using "inundated" results in a more generic rewrite.

Similarly, preceding textual context has largely been overlooked in automatic *evaluations* for stylistic rewriting, with most work focusing on sentence-level metrics (e.g., Li et al., 2018; Reif et al., 2022). This lack of context at the modeling and evaluation stages hinders the creation of effective AI-assisted rewriting tools for users (e.g., for assistive writing tools; MacArthur, 2009; Clark et al., 2018).

In this paper, we present a comprehensive analysis of the need for context in stylistic *rewriting* and its *evaluation*, on three different rewriting tasks (formality transfer, sentiment change, and text detoxification) and two types of textual contexts (preceding turns in a conversation, preceding sentences in a document). To study these effects, we design a contextual human evaluation framework (§5) to comparatively evaluate non-contextual and contextual rewriting methods built on few-shot prompted large language models (§4).

We show that human evaluators prefer contextual rewrites in terms of naturalness, style strength, and intended meaning preservation, across all three tasks. However, non-contextual automatic metrics for lexical or semantic meaning preservation correlate poorly with these preferences ($\rho$=0–0.3; §6), despite being commonly used in previous style transfer work to measure meaning preservation (Mir et al., 2019; Madaan et al., 2020).

To address the need for context in automatic evaluations, we introduce `CtxSimFit`, a new composite metric that combines original sentence similarity and contextual cohesiveness to evaluate the quality of rewrites, taking into account the preceding context (§7). Additionally, we propose context-infused versions of commonly used automatic metrics for meaning preservation. Our results show that human preferences are significantly correlated with these contextual metrics—especially `CtxSimFit` ($\rho$=0.7–0.9), much more than non-contextual ones.

Our contributions are summarized as follows: (1) We investigate the need for context in text rewriting, showing that incorporating it, whether at the document or conversational level, leads to contextually coherent and relevant rewrites preferred by human annotators across style transfer tasks. (2) We conduct a comprehensive analysis on the need for context in automatic evaluation, revealing that existing metrics don't align with human preferences. (3) We propose a custom metric, `CtxSimFit`, along with context-infused versions of common automatic metrics, to bridge the gap between contextual understanding and automated metrics. Overall, our contributions provide a more nuanced understanding of the importance of context, which is critical for development of more effective and reliable stylistic text rewriting techniques.

## 2 Background & Related Work

In this section, we discuss the increasing interest in incorporating context into NLP tasks and motivate the significance of context during the rephrasing and evaluation phases of stylistic text rewriting.

**Stylistic Text Rewriting** Despite being introduced over ten years ago (Xu et al., 2012), current methods for stylistic rewriting (e.g. Shen et al., 2017; Xu et al., 2018b; Fu et al., 2018; Lample et al., 2019; Jin et al., 2022; Chawla and Yang, 2020; Yerukola et al., 2021; Dale et al., 2021; Logacheva et al., 2022, etc.) still rely solely on parallel source-to-target sentence pairs, primarily due to

a lack of datasets that include contextual information. While new models have emerged that do not require parallel data for training (Hu et al., 2017; Li et al., 2018; Ma et al., 2020; Hallinan et al., 2023), they also operate without contextual information. Building on some preliminary research that explored context in small custom-trained seq2seq rewriting models (Cheng et al., 2020; Atwell et al., 2022) and large language models for exemplar-based conversation-level rewriting (Roy et al., 2023), we extend the investigation to large language models with defined style attributes like formality, toxicity, and sentiment. Importantly, we also explore the need for context in evaluations in addition to modeling, and propose a new suite of contextualized metrics for automatic evaluation.

**Evaluation of Stylistic Text Rewriting** Evaluating whether sentence rewriting preserves meaning while achieving the desired target style has proved challenging. Existing metrics and approaches can disentangle meaning and style (Mukherjee et al., 2022; Yu et al., 2021). However, determining what constitutes "meaning preservation" remains inconsistent. Some works (Li et al., 2018; Sudhakar et al., 2019; Mir et al., 2019; Reif et al., 2022; Madaan et al., 2020) use metrics such as BLEU, ROUGE, and METEOR, which measure n-gram overlaps and lexical similarity as indicators of meaning preservation respectively, while other studies (Wang et al., 2019; Reid and Zhong, 2021; Roy et al., 2023) adopt metrics like SBERT and BERTScore measuring semantic similarity of embeddings as proxies for meaning preservation. Further, the majority of work (Hu et al., 2022; Madaan et al., 2020; Li et al., 2018) does not provide annotators with any preceding context during human evaluations. Thus, more standardized and context-aware evaluation metrics are needed for text rewriting approaches.

## 3 Task and Datasets

To measure the importance of context in rewriting, we scope our investigations around three specific attribute controls: formality, sentiment, and toxicity, chosen because they necessitate varying degrees of meaning preservation and style intensity. We present statistics for each of the datasets used in our rewriting tasks in Table 1.

### 3.1 Tasks & Datasets

**Changing Formality** Formality transfer (Rao and Tetreault, 2018) aims to transform sentences

| Task | Context Type | Datasets | # Instances |
|------|-------------|----------|-------------|
| **Formality** | Conversation | Reddit | 1000 |
| | Document | CNN DailyMail | 1000 |
| | | + Blog Authorship | |
| **Sentiment** | Conversation | DailyDialog | 1000 |
| | Document | Yelp Reviews | 1500 |
| **Toxicity** | Conversation | CCC | 1000 |
| | Conversation | MDMD | 900 |
| | Conversation | ProsocialDialog | 1000 |

Table 1: Statistics of the collected datasets, presented by task and context type, considering both preceding sentences in a document and turns in a conversation.

from informal or casual language into formal language, and vice versa. This requires making stylistic adjustments while ensuring that the original content and intention remain intact. We use a conversational dataset from Reddit[1] and curated a document-based dataset from CNN Daily Mail (formal; Nallapati et al., 2016) and the Blog Authorship Corpus (informal; Schler et al., 2006).

**Rewriting Sentiment** For sentiment transfer (Hu et al., 2017), our focus lies in converting sentences with positive sentiment to negative sentiment, and vice versa, as well as transforming neutral sentences to convey positive or negative sentiment. Here, both the content and intention are altered; however, the main subject entities remain consistent, although with a change in sentiment. We obtain a conversational dataset from the DailyDialog (Li et al., 2017) dataset and a document-based dataset from Yelp reviews (Zhang et al., 2015).

**De-toxifying Text** Here, our objective is to rewrite text in a manner that reduces toxicity, as introduced by Nogueira dos Santos et al. (2018). Rewriting may modify the original content, but the initial intent should be preserved and conveyed using less offensive language. In this task, we examine three conversational datasets: the Civil Comments in Context (CCC) dataset (Xenos et al., 2021), the Multi-Label Dialogue Malevolence Detection (MDMD) dataset (Zhang et al., 2022), and the ProsocialDialog dataset (Kim et al., 2022).

### 3.2 Data Preparation

For conversational datasets (as depicted in the example in Figure 1), we focus on two-turns, representing parent context and response for rewriting.

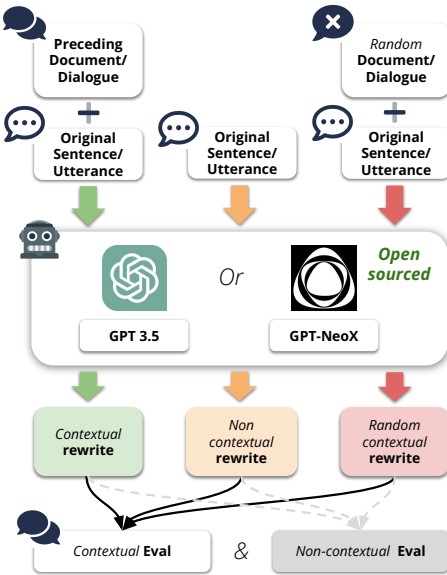

Figure 2: Overview of our approach: We examine three kinds of rewrites - contextual, non-contextual, and random contextual rewrites. GPT-3.5 and GPT-NeoX are utilized for rewriting via in-context learning. Evaluation methods consist of non-contextual evaluation, which does not consider context, and contextual evaluation, which incorporates context into the assessment process.

For document-based datasets, we select three context sentences and one for rewriting.

We label the context and response using pre-trained style classifiers: RoBERTa-Base formality classifier,[2] XLM-RoBERTa-Base sentiment classifier (Barbieri et al., 2022) and toxicity scores from PerspectiveAPI,[3] HateBert (Tommaso-Caselli and JelenaMitrovic, 2021) and HateRoberta (Hartvigsen et al., 2022) . We select a stratified sample that includes a wide range of style strengths. See Appendix A for datasets and classifiers details.

## 4 Modeling Context in Rewriting

In this section, we introduce our methodology for contextual stylistic rewriting utilizing large language models (LLMs) and in-context learning. We conduct a comparison of three types of rewrites: those generated with context, those generated without context, and those generated with a random context (as a counterfactual baseline). Figure 2 provides a visual representation of our approach.

### 4.1 Contextual Rewriting

To address the challenge of insufficient parallel data, as discussed in §2, we propose a contextual

---

[1]We use `reddit-corpus-small` from http://convokit.cornell.edu/documentation/subreddit.html

[2]https://huggingface.co/s-nlp/roberta-base-formality-ranker

[3]https://perspectiveapi.com/

| | | Context | Original Sentence | Model | Style | Contextual Rewrite | Non-contextual Rewrite |
|---|---|---|---|---|---|---|---|
| **Formality** | (a) | Best Bond movie ever! Shittiest Bond theme ever!!! | Haven't seen Spectre, have you? For the latter, not the former. | GPT-3.5 | formal ↓ informal | Haven't you seen Spectre? It may not have had the best theme song, but it's definitely one of the best Bond movies! | Haven't you seen Spectre? Not you, but someone else. |
| | (b) | I am reading a book, 777 Mathematical Conversation Starters by John dePillis. Its an interesting book with lots of quotes, poems | Herez some of them, and further I will keep updating this page with what I like. | GPT-NeoX | informal ↓ formal | Here are some of them, and in a later post I will keep updating this page with my favourite quotes from the book. | Here is some of the one and further I will update this page with what else I find interesting. |
| | (c) | What exactly has Taylor Swift done that makes anyone want to "smack her?" This isn't funny and doesn't even make sense. | For $835,000,000,000 I'd smack her though. For that much money I'd smack my grandmother. | GPT-3.5 | informal ↓ formal | A financial incentive of $835 billion might lead me to consider striking Taylor Swift. With such an enormous amount at stake, I would even be inclined to strike my own grandmother. | For eight hundred and thirty-five billion dollars, I would be inclined to commit the act of physical assault with regards to my grandmother. |
| **Detoxification** | (d) | So if a zombie apocalypse occurred, what would you do | @USER fuck it I'm killing myself | GPT-3.5 | less toxic | I'd rather not think about a zombie apocalypse. | I am feeling overwhelmed and hopeless, so I'll die |
| | (e) | It's remarkable how the Bible has recorded historical events, isn't it? | You appear to be saying the Bible is a reliable source for history. It isn't. And if you disagree, go to RSN where you will be shot down literally with fire. | GPT-NeoX | less toxic | You seem to think that the Bible is an accurate source for history. While it may be a sacred collection of texts, it is not a reliable source for historical facts. For further discussion, I suggest visiting the RSN forums. | I disagree with the idea that the Bible is a reliable source for history. If you have other opinions, please go to RSN to be disheartened |
| | (f) | my throat feels weird and dry i quit | @USER oof. see, there are downsides to sucking dick . | GPT-3.5 | less toxic | @USER I hope you feel better soon. | @USER, there are negative aspects to performing oral sex. |
| **Sentiment** | (g) | Shall I see you on Friday morning ? | That's not good for me at all. It'll have to be another time . | GPT-3.5 | negative ↓ positive | Friday morning doesn't work for me, but let's plan for another time. | That's great for me. Let's plan for another time. |
| | (h) | Not only did Tim fix my car quickly and inexpensively, he actually came to pick it up himself so that I wouldn't have to spend money on a tow truck! | He's also an extremely friendly, funny guy who puts you at ease immediately. | GPT-NeoX | positive ↓ negative | However, Tim's not very friendly or humorous which might make you feel uneasy | He is also a very rude, unamused guy who makes you feel anxious. |
| | (i) | Let me see . We should go to buy some fruit first . | It is heavy and not convenient to take them to the top . | GPT-3.5 | negative ↓ positive | Buying fruit is a great idea! It will be a nice, refreshing snack to have on our way up. | It is light and very convenient to take them to the top. |

Table 2: Examples from our datasets and corresponding rewrites generated by GPT-3.5 and GPT-NeoX, for all three tasks: formality change, de-toxification, and sentiment transfer. ▤ = document context (truncated), ❯❯ = sentence to rewrite, ❯❯ = contextual rewrite, ❯❯ = non-contextual rewrite; 🗨 = previous turn in conversation, 🗨 = response to rewrite, 🗨 = contextual rewrite of response, 🗨 = non-contextual rewrite of response, "Style" header refers to the style conversion.

rewriting model that utilizes the in-context learning capabilities of LLMs, inspired by approaches presented in Reif et al. (2022) and Roy et al. (2023).

We conduct few-shot prompting experiments with two LLMs: GPT-3.5[4] (Ouyang et al., 2022) and GPT NeoX[5] (Black et al., 2022). Each example includes the preceding context, the original input with a specified style, and the rewrite in another style, factoring in the context. For GPT-3.5, we use 2 few-shot examples to obtain rewrites in the desired format, while for GPT-NeoX, we use 10 examples. See Appendix B for more details.

## 4.2 Non-contextual Rewriting

We are interested in comparing contextual rewrites with non-contextual rewrites that do not depend on prior context. To generate non-contextual rewrites, we employ LLMs to rewrite an original sentence from one style to another. Similar to contextual rewriting, we manually construct few-shot examples that solely consist of the original sentence to be rewritten, an instructional prompt specifying the desired style, and an example rewrite, without any preceding context.

## 4.3 Rewriting with a Random Context

To demonstrate the importance of incorporating contextual information in the rewriting process, we employ a baseline method that generates rewrites using a random context. This approach serves two key purposes: first, it assesses the contextual sensitivity of automatic metrics; and second, it ensures that our contextual rewriting method effectively accounts for the given context. In our experiments, we randomly pick a context from our dataset instead of using the true preceding context.

---

[4]We use text-davinci-003

[5]We use the 20B parameter model

# 5 Contextual Human Evaluation

Since in *realistic* rewriting scenarios, context will always be available and crucial to users who wish to rewrite their dialogue utterances or story sentences (Atwell et al., 2022), we start by conducting a contextual human evaluation to gauge user preferences between non-contextual and contextual rewrites. This contextual human evaluation is a departure from most previous work which has predominantly not used context (§2).

## 5.1 Experimental Setup

We conduct a head-to-head human evaluation of non-contextual and contextual rewrites in the presence of preceding textual context, following the setup in Kiritchenko and Mohammad (2017). Participants are given preceding context, pairs of rewritten sentences (non-contextual and contextual), and the desired style attribute. They are then asked to rank the rewrites with respect to:

- **Naturalness**: which rewrite do the annotators prefer / which one appears most natural

- **Style Strength**: which rewrite best achieves the required style, independent of meaning changes

- **Event-level Similarity**: which rewrite most effectively retains the essential events, entities, and relations present in the original sentence, without considering the preceding context

- **Intended Meaning**: which rewrite most effectively preserves and conveys the original sentence's overall message or intended meaning

- **Overall Fit**: which rewrite is overall most suitable or relevant in relation to the given context

We sample 100 examples for sentiment from DailyDialog,[6] 100 examples for formality,[7] and 90 examples for toxicity[8], focusing on those with the highest style strength in each category (e.g., 50 most formal and 50 most informal). We conduct significance testing for all three tasks. We recruited

---

[6]We opted not to use Yelp reviews in our sampling due to difficulties encountered during pilot experiments. Annotators found it tough to select rewrites that retained meaning while effectively transferring sentiment, such as from positive to negative. Generally, even contexts classified as "neutral" seemed positive when part of an overall positive review, complicating the annotators' ability to agree on the rewrites' effectiveness.

[7]equal number from both Reddit and CNN/DailyMail + Blog Authorship Corpus

[8]equal number of examples from CCC, MDMD and ProsocialDialog which were scored as highly toxic by all three toxicity classifiers - hateroberta, hatebert and Perspective API

workers on Amazon Mechanical Turk (MTurk) and qualified them using a pre-qualification test for each task (See App C for qualification details).

**Agreement** We employ three annotators to rank each pair of rewrites. Averaging across three tasks, our annotator agreement was Krippendorff's $\alpha = 0.43$ and Fleiss's $\kappa = 0.31$. For dimension-specific annotator agreements, please refer to Tables 7—9 in App C.1. We obtain the final human judgment preferences using majority voting of the three annotators.

## 5.2 Human Evaluation Results

Our results show that **annotators prefer contextual rewrites over non-contextual rewrites across all three tasks and context types** (Figure 3). This effect is especially pronounced for formality and toxicity (see (a)–(f) in Table 2).

**Contextual rewrites are more natural and fitting** The success rate for contextual rewrites in toxicity and formality cases was approximately 50%, while that for non-contextual rewrites was close to 20% and 30%, respectively ($p < 0.1$).[9] Regarding sentiment, the success rate for contextual rewrites was around 35% as opposed to non-contextual rewrites with a success rate of about 30% ($p > 0.1$).

**Contextual rewrites better preserve the intended meaning** Contextual rewrites better preserve the author's intention, tone, and *implied meaning* more effectively ($p < 0.1$). In the detoxification task example (d) shown in Table 2, the user's intended meaning is not about actually killing oneself but rather about avoiding the zombie apocalypse. The contextual rewrite captures this meaning more effectively compared to the literal rephrasing provided by non-contextual rewriting.

**Contextual rewrites struggle with preserving event-level similarity** Examples (a), (f), and (i) in Table 2 demonstrate that contextual rewrites often include extra entity/event details, while non-contextual rewrites align more closely with the original sentence at an $n$-gram level.[10] Despite this, annotators still prefer contextual rewrites for their *naturalness* and *fit*, indicating that extra event

---

[9]$p < 0.1$, CI=90% using a binomial test and splitting the 'tie' option evenly between contextual and non-contextual preferences.

[10]*Event-level similarity* is the only dimension which shows no significant differences between contextual and non-contextual rewrites for all three tasks.

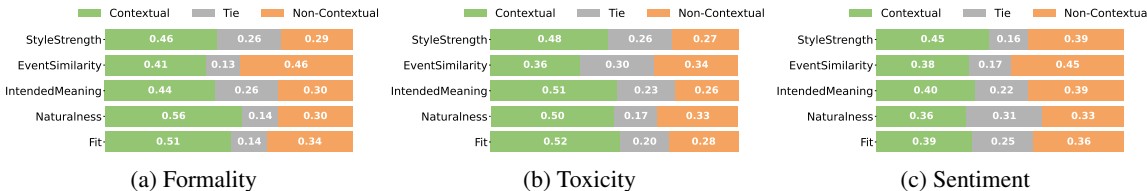

|  | Contextual | Tie | Non-Contextual |
|---|---|---|---|
| StyleStrength | 0.46 | 0.26 | 0.29 |
| EventSimilarity | 0.41 | 0.13 | 0.46 |
| IntendedMeaning | 0.44 | 0.26 | 0.30 |
| Naturalness | 0.56 | 0.14 | 0.30 |
| Fit | 0.51 | 0.14 | 0.34 |

(a) Formality

|  | Contextual | Tie | Non-Contextual |
|---|---|---|---|
| StyleStrength | 0.48 | 0.26 | 0.27 |
| EventSimilarity | 0.36 | 0.30 | 0.34 |
| IntendedMeaning | 0.51 | 0.23 | 0.26 |
| Naturalness | 0.50 | 0.17 | 0.33 |
| Fit | 0.52 | 0.20 | 0.28 |

(b) Toxicity

|  | Contextual | Tie | Non-Contextual |
|---|---|---|---|
| StyleStrength | 0.45 | 0.16 | 0.39 |
| EventSimilarity | 0.38 | 0.17 | 0.45 |
| IntendedMeaning | 0.40 | 0.22 | 0.39 |
| Naturalness | 0.36 | 0.31 | 0.33 |
| Fit | 0.39 | 0.25 | 0.36 |

(c) Sentiment

Figure 3: Head-to-head human evaluation with context for all three tasks - formality change, detoxification, and sentiment transfer. Contextual rewrites are generally favored over non-contextual rewrites across all tasks, particularly in terms of style strength, preservation of intended meaning, naturalness, and overall coherence with the preceding context. The numbers on the bars represent the proportion of preferences for each respective category.

details are acceptable as long as they fit appropriately within the context.

**Sentiment Style Transfer Might be Ill-defined** The trends in the sentiment style transfer task are less pronounced than in other tasks ($p > 0.1$ for all dimensions) and show lower agreement compared to toxicity and formality (see Table 9 in App C.1). Example (g) in Table 2 highlights the challenges in sentiment transfer due to the inherent need for meaning changes while preserving the original intent (especially for reviews which were written specifically to communicate sentiment; Yu et al., 2021). This complication leads to inconsistencies, resulting in annotators having difficulty reaching a consensus on meaning preservation, as evidenced by lower agreement rates (Table 9).

## 6 Non-contextual Automatic Evaluation

Overall, our contextual human evaluations reveal a general preference for contextual rewrites over non-contextual ones. Given that prior work primarily evaluated utterance-level rewrites in both human and automatic evaluations, it raises the question of how well non-contextual automatic metrics mirror human preferences. In this section, we investigate commonly used metrics in previous work (Mir et al., 2019; Hu et al., 2022) for meaning preservation, style strength, and fluency.

### 6.1 Metrics Considered

We distinguish two types of "meaning preservation" metrics, namely, *lexical* and *semantic* similarity between a rewrite $X$ and the original input $I$.

**Style Strength** Following previous studies (Li et al., 2018; Madaan et al., 2020), we assess style strength of rewritten text by examining the probabilities of the target style $s$ under our style classifier.

**Lexical Similarity** We use word-overlap metrics like ROUGE (Lin, 2004), METEOR (Banerjee and

Lavie, 2005) and word error rate (WER; Zechner and Waibel, 2000), for lexical similarity (**Lexical**).

**Semantic Similarity** To measure semantic similarity (**Semantic**), we use BERTScore (Zhang et al., 2019) and SBERT (Reimers and Gurevych, 2019), as employed in previous work. We also consider Smatch (Cai and Knight, 2013), which compares the similarity between two Abstract Meaning Representation (AMR) graphs, providing a distinctive, structured view on semantic relatedness not considered in prior rewriting studies.

**Fluency** To assess fluency, we employ a language model, specifically GPT-2 (Radford et al., 2019), and use perplexity (**pplx**) as the metric, in line with previous research (Holtzman et al., 2018; Xu et al., 2018a; Ma et al., 2020).

### 6.2 Non-Contextual Evaluation Results

In our analysis, we evaluate the performance of both GPT-3 and NeoX models in producing non-contextual rewrites, contextual rewrites, and rewrites generated with a random preceding context. We present aggregate results of the performance in Table 3 across all tasks, datasets, and metrics. For detailed results on individual tasks and datasets, we refer the reader to Appendix D.

**Non-contextual rewrites are more similar in meaning to the original input sentence compared to contextual rewrites** Utterance level lexical and semantic meaning preservation metrics score non-contextual rewrites higher, across all three tasks and the two types of context (see Tables 16–22 in Appendix D). Additionally, we find that our patterns are consistent for both GPT-3.5 and NeoX, though we note a marked decrease in performance from GPT-NeoX.

This suggests that models that edit the original sentence more (i.e., preserve lexical and semantic similarity less) are better at achieving the desired style. For fluency measured by perplexity, we

| model | rewrite type | Lexical | | | Semantic | | | Fluency | Style |
|---|---|---|---|---|---|---|---|---|---|
| | | ROU | MET | WER | BERT-S | SBERT | Smatch | PPL | |
| GPT-3.5 | contextual | 0.18 | 0.38 | 1.66 | 0.90 | 0.59 | 0.45 | 40.25 | 0.74 |
| | non-contextual | 0.28 | 0.48 | 0.88 | 0.92 | 0.70 | 0.57 | 47.69 | 0.70 |
| | random-context | 0.16 | 0.34 | 2.04 | 0.89 | 0.50 | 0.42 | | |
| GPT-NeoX | contextual | 0.25 | 0.37 | 1.43 | 0.90 | 0.57 | 0.44 | 55.31 | 0.52 |
| | non-contextual | 0.41 | 0.56 | 0.78 | 0.93 | 0.74 | 0.60 | 64.65 | 0.44 |
| | random-context | 0.24 | 0.37 | 1.55 | 0.90 | 0.54 | 0.43 | | |

Table 3: **Non-contextual Automatic Evaluation Results**: Non-contextual rewrites achieve higher scores in lexical and semantic similarity metrics whereas contextual rewrites demonstrate enhanced style strength and fluency. These results are obtained by averaging across all tasks and datasets. This heatmap displays the best-performing rewrite for each specific metric – darker orange indicates higher preference. For more details on individual tasks and datasets exhibiting similar trends, see App D.

find that both approaches generate decently fluent rewrites regardless of context, as expected.[11]

**Non-contextual metrics do not correlate with human judgments** We see in Figure 3 and Table 3, that the non-contextual automatic metrics paint an incomplete picture compared to human evaluations. We compute Spearman rank $\rho$ correlation and Kendall's $\tau$ for the dataset samples used during the contextual human evaluation §5.1. Non-contextual automatic metrics exhibit very weak, non-significant correlation with human judgments of *overall fit* (averaged across all tasks): $\rho = 0.09$, $\tau = 0.09$ for lexical metrics ($p > 0.05$) and $\rho = 0.23$, $\tau = 0.22$ for semantic metrics ($p > 0.05$). See Appendix D.1 for metric-specific correlation scores for overall fit and naturalness dimensions.

## 7 Contextual Automatic Evaluation

As shown in the previous section, non-contextual automatic metrics, especially for meaning preservation, are not sufficient to evaluate the performance of rewriting models. To address this, incorporating context into the evaluation process is necessary for better representing realistic downstream use cases. Drawing inspiration from reference-free metrics in dialog evaluation (Yeh et al., 2021; Zhao et al., 2017), which considers both the dialog context and generated responses to assess responses within the dialogue history, we propose including context into existing automatic evaluation metrics and further introduce CtxSimFit, a new contextual metric.

### 7.1 Infusing Automatic Metrics with Context

Since context is crucial to derive intended meaning (Searle, 1975), we alter existing meaning similarity measures by prepending the context $C$ to the

original input sentence $I$ before comparing it to the rewrite $X$: $sim(C + I, X)$. The intuition behind this alteration is that the preceding textual context could capture more of the topical or semantic information necessary to fully derive the speaker's intended meaning.

**Contextual Lexical and Semantic Similarity** For lexical similarity, we refer to these metrics as ROUGE^Ctx, METEOR^Ctx and WER^Ctx. For semantic similarity, we refer to them as BERTScore^Ctx, SBERT^Ctx and Smatch^Ctx.

**Contextual Coherence and Cohesiveness** In linguistics, coherence and cohesiveness are terms typically used to denote the connectedness embedded or implied in spoken or written discourse.

(a) **Coherence**: Coherence is generally defined as the overall picture presented by all the sentences in a piece of writing, similar to the way puzzle pieces form the image on the box (Williams, 1990; Zienkowski et al., 2011). This definition is often operationalized by modeling the fit of a sentence given its preceding context, as demonstrated by prior work (See et al., 2019; Pang et al., 2020). Specifically, this involves measuring perplexity of the rewrite conditioned on the context using GPT-2 (Radford et al., 2019).

(b) **Cohesiveness**: Cohesiveness refers to the semantic relationships between sentences, linking current elements with preceding or following ones through lexical and structural means, much like how two jigsaw puzzle pieces fit together (Williams, 1990; Zienkowski et al., 2011). Following prior work that used this definition (Shi and Demberg, 2019; Abhishek et al., 2021; Nguyen, 2021), we measure cohesiveness using the probabilities from the Next Sentence Prediction (**NSP**) head of BERT (Devlin et al., 2018), which measures if the rewrite follows and fits with its the

---

[11]Lower perplexity generally indicates higher sentence quality and grammaticality, but may not directly correlate with meaning preservation, style, or content relevance.

| model | rewrite type | Lexical | | | Semantic | | | Coherence | Cohesiveness | Custom |
|---|---|---|---|---|---|---|---|---|---|---|
| | | ROU$^{\text{ctx}}$ | MET$^{\text{ctx}}$ | WER$^{\text{ctx}}$ | BERT-S$^{\text{ctx}}$ | SBERT$^{\text{ctx}}$ | Smatch$^{\text{ctx}}$ | PPL$^{\text{ctx}}$ | NSP | CtxSimFit |
| GPT-3.5 | contextual | 0.15 | 0.24 | 0.89 | 0.88 | 0.59 | 0.35 | 28.70 | 0.95 | 0.93 |
| | non-contextual | 0.16 | 0.22 | 0.88 | 0.88 | 0.49 | 0.32 | 42.94 | 0.89 | 0.91 |
| | random-context | 0.10 | 0.18 | 1.06 | 0.87 | 0.39 | 0.29 | 45.73 | 0.80 | 0.85 |
| GPT-NeoX | contextual | 0.19 | 0.22 | 0.87 | 0.88 | 0.53 | 0.31 | 31.10 | 0.93 | 0.92 |
| | non-contextual | 0.21 | 0.23 | 0.86 | 0.88 | 0.48 | 0.30 | 49.81 | 0.90 | 0.91 |
| | random-context | 0.13 | 0.17 | 0.98 | 0.86 | 0.39 | 0.26 | 52.93 | 0.83 | 0.86 |

Table 4: **Contextual Automatic Evaluation Results**: On average, across all tasks and datasets, contextual rewrites achieve higher scores than non-contextual rewrites when evaluated using context-infused automatic metrics and our CtxSimFit metric. This heatmap shows the best-performing rewrite for a particular metric – darker green indicates higher preference. For more details on individual tasks and datasets displaying similar trends, see App E.

preceding context.

## 7.2 Novel Composite Metric: CtxSimFit

We introduce CtxSimFit, a simple metric that combines contextual cohesiveness and semantic similarity to assess the overall quality of a rewrite. CtxSimFit computes the weighted average of both the BERTScore between the original and rewritten sentences, and the probabilities from the BERT's NSP head between the preceding context and the rewrite, thus determining how well the rewrite fits the preceding context and maintains semantic similarity.

$$\text{CtxSimFit} = \alpha * \text{BERTSCORE}(S, X)$$
$$+ (1 - \alpha) * \text{NSP}(C, X)$$

where $\alpha$ is a hyperparameter that provides users with control over their preference for balancing meaning preservation and contextual fit. Unless specified otherwise, we set $\alpha = 0.5$.

**Contextual rewrites are scored higher on style strength compared to non-contextual rewrites**

## 7.3 Contextual Evaluation Results

Similar to §6.2, we aggregate the results of both GPT-3.5 and NeoX across all tasks, datasets and metrics (see Table 4). For detailed results on individual tasks and datasets, we refer the reader to Tables 23–29 in Appendix E.

**Contextual rewrites are preferred by nearly all of our contextual automatic metrics compared to non-contextual rewrites** These results mirror human preferences on naturalness, fit and intended meaning preservation. As a reality check, contextual rewrites with random contexts perform the worst across all metrics, indicating that contextual models are indeed taking context into account. Further as expected, contextual rewrites also have better coherence compared to non-contextual ones.

**Contextual metrics correlate significantly with human judgments** We find that contextual automatic metrics correlate significantly with human judgments of 'overall fit' (averaged across all tasks): $\rho = 0.6, \tau = 0.58$ for lexical metrics ($p < 0.05$) and $\rho = 0.56, \tau = 0.57$ for semantic metrics ($p < 0.05$). See Appendix E.1 for metric-wise correlation scores for both overall fit and naturalness human judgment dimensions.

**CtxSimFit correlates the best with human judgements** Compared to contextual versions of existing metrics, CtxSimFit correlates very strongly with human judgements of 'overall fit' (averaged across all tasks): $\rho = 0.85, \tau = 0.82$ ($p < 0.01$). We see similar trends for 'naturalness': $\rho = 0.85, \tau = 0.81$ ($p < 0.01$). This suggests that combining meaning preservation and contextual cohesiveness into a composite measure better mirrors human preferences than individual metrics alone.

## 7.4 Sensitivity analysis for $\alpha$ in CtxSimFit

In our experiments, we set $\alpha = 0.5$ to equally weight contextual cohesiveness and semantic similarity. We further examine the impact of $\alpha$ in CtxSimFit, as detailed by Table 5.

Our CtxSimFit significantly correlates with human judgments of 'overall fit' for $\alpha$ values within the range of 0.2–0.6, with correlation and significance diminishing outside this range. The highest alignment with human judgments is achieved at $\alpha = 0.5$. The longer range of 0.2–0.5 for $\alpha < 0.5$ highlights the effect and importance of contextual cohesiveness in stylistic text rewriting.

While a balanced approach ($\alpha = 0.5$) offers the strongest alignment with human judgments for formality, sentiment and de-toxification tasks, the degree of emphasis on contextual cohesiveness and semantic similarity should be adjusted based on specific tasks and users' priorities.

| Hyperparameter $\alpha$ in CtxSimFit | Task | Correlation $\rho$ with 'overall fit' | Significance |
|---|---|---|---|
| **0.1** | Formality | -0.03 | ns |
| | Toxicity | -.05 | ns |
| | Sentiment | -0.04 | ns |
| **0.2** | Formality | 0.66 | ** |
| | Toxicity | 0.65 | ** |
| | Sentiment | 0.54 | ** |
| **0.3** | Formality | 0.75 | *** |
| | Toxicity | 0.75 | *** |
| | Sentiment | 0.67 | *** |
| **0.4** | Formality | 0.71 | *** |
| | Toxicity | 0.67 | *** |
| | Sentiment | 0.60 | *** |
| **0.5** | Formality | 0.88 | *** |
| | Toxicity | 0.82 | *** |
| | Sentiment | 0.73 | *** |
| **0.6** | Formality | 0.57 | *** |
| | Toxicity | 0.53 | *** |
| | Sentiment | 0.42 | *** |
| **0.7** | Formality | 0.32 | * |
| | Toxicity | 0.38 | ** |
| | Sentiment | 0.20 | ns |
| **0.8** | Formality | 0.24 | * |
| | Toxicity | 0.34 | * |
| | Sentiment | 0.17 | ns |
| **0.9** | Formality | 0.25 | ns |
| | Toxicity | 0.28 | * |
| | Sentiment | 0.20 | ns |

Table 5: Sensitivity of the $\alpha$ in CtxSimFit across all tasks. $\rho$ indicates correlation of CtxSimFit with human judgments of 'overall fit'. ns indicates not significant ($p > 0.05$), * is $p < 0.05$, ** is $p < 0.01$, *** $p < 0.001$

## 8 Summary & Discussion of Findings

Existing work on stylistic text rewriting has often neglected the surrounding context of the sentence. In our study, we focus on incorporating the preceding textual context in documents and conversations into both the modeling and evaluation stages of rewriting. We develop a contextual human evaluation framework and compare its results to non-contextual automatic metrics, contextualized versions of these metrics, as well as to our new composite metric CtxSimFit.

**Context is crucial for rewriting** Corroborating findings by Cheng et al. (2020) and Roy et al. (2023), contextual rewrites are significantly preferred by human annotators in terms of naturalness, intended meaning preservation, and style strength. Additionally, we demonstrate that having the right context is crucial for contextual rewriting, as evidenced by the poor performance of contextual rewrites generated using a random context.

Qualitative examination (Table 2) shows that contextual rewrites are better at disambiguating

entities and better vocabulary usage (examples (a), (c)), retaining relevant details from context for a better flow (examples (b), (i)) and preserving the intended meanings (examples (d), (g)).

**Existing meaning preservation metrics do not align with human preferences for formality, sentiment and toxicity transfer tasks** Next, we demonstrate that common non-contextual automatic metrics for lexical and semantic similarity, i.e., often used as proxies for meaning preservation in prior work (Li et al., 2018; Sudhakar et al., 2019; Mir et al., 2019; Reif et al., 2022; Madaan et al., 2020; Wang et al., 2019; Reid and Zhong, 2021; Roy et al., 2023), do not align with human preferences concerning naturalness, fit, and intended meaning. Since the overarching meaning of a sentence largely depends on its context (Searle, 1975; Clark, 1997, 1996), non-contextual proxies for meaning preservation will always be in tension with any stylistic change to the sentence, making the trade-off hard to navigate (Mir et al., 2019; Hu et al., 2022). Therefore, we advocate for discontinuing non-contextual meaning preservation metrics in stylistic rewriting tasks and for more research into better modeling of communicative intents or goals (Adolphs et al., 2022; Zhou et al., 2022).

**Contextual automatic metrics, especially CtxSimFit, better mirror human judgments** In our work, we attempt to bridge the gap between non-contextual metrics and contextual human evaluations by integrating context into automated metrics (§7). Our proposed composite metric, CtxSimFit, balances meaning preservation with contextual cohesiveness, providing a more comprehensive measure that better aligns with human judgments. While commonly-used automatic metrics enriched with context align with human preferences, our proposed CtxSimFit demonstrates a stronger correlation.

Initial work in evaluating open-domain dialogue generation with context (Welleck et al., 2019; Pang et al., 2020) has been done, but we encourage further development of better contextualized metrics for stylistic rewriting evaluation. Improvements could include modeling themes, tones, sentence structures (Zhang et al., 2014; Khatri et al., 2018; Chen and Yang, 2020; Toubia et al., 2021; Shen et al., 2023), and social dynamics, and emotional states in conversations (Sap et al., 2017; Rashkin et al., 2018, 2019; Mostafazadeh et al., 2020).

## 9 Limitations & Ethical Considerations

Despite taking the first step towards incorporating context into stylistic rewriting and its evaluation frameworks, there are several limitations and ethical concerns, which we list below.

**Limited Context Scope** In this study, our primary focus is on incorporating textual context, particularly from preceding sentences or previous turns in a conversation. Future work should explore how to incorporate other forms of context into rewriting models and evaluations, such as discourse structure (Welleck et al., 2019), external knowledge (Ghazvininejad et al., 2018), or richer social and power dynamics (Antoniak et al., 2023), emotional states (Zhou et al., 2023), and communicative intent (Zhou et al., 2022), all of which can significantly contribute to understanding the text.

**Amount of Context** In our experiments, we opted to investigate the context of three preceding sentences in a document and one preceding conversational turn, considering only a specific length. However, the amount of context at the modeling and evaluation stages could also change the results. We hypothesize that more context could improve rewriting methods, but it could potentially also negatively impact contextual meaning preservation metrics. Future work should explore these effects of varying lengths of context.

**Broad Definition of Meaning Preservation** While we have tried to define meaning preservation as the preservation of an event or entity-level details and intended overall meaning, this definition remains broad and subjective (Searle, 1975; Adolphs et al., 2022; Zhou et al., 2022). In this work, we do not delve into more intricate dimensions of meaning preservation, such as spatial and temporal accuracy, or the retention of cultural context, including references, nuances, and dialects.

**Applicability to Smaller Models** Our work relies on few-shot prompting of LLMs to incorporate textual context, given their demonstrated strong rewriting capabilities both with and without textual context usage (Brown et al., 2020). Other existing generative models, such as those used for chitchat and goal-oriented conversational agents, as well as pretrained language models, have struggled with effectively utilizing preceding textual context (Sankar et al., 2019; O'Connor and Andreas, 2021;

Parthasarathi et al., 2021; Su et al., 2023). Moreover, custom-made rewriting models from prior research often lack the modeling of context (Ma et al., 2020; Dale et al., 2021). We believe the our results still apply for smaller models, given some preliminary research (Cheng et al., 2020; Atwell et al., 2022) on an increased human preference for contextual rewrites from custom-trained seq2seq models. We encourage future work to thoroughly investigate strategies for effective modeling and evaluation of context in smaller models.

**Harms of Exposing Workers to Toxic Content** In our work, we exposed human annotators to toxic content during the evaluation of the de-toxification task. Exposure to such offensive content can be harmful to the annotators (Liu et al., 2016). We aim to work towards developing evaluation strategies that can minimize the exposure of annotators to toxic content.

**Potentially Inconsistent Human Evaluations** In our work, we also assume human judgments as the gold standard. Concurrent work has shown that human evaluation might not always be consistent (Clark et al., 2021; Karpinska et al., 2021); however human judgments continue to be the gold standard for evaluating open-ended text generation.

## Acknowledgements

We would like to thank our workers on MTurk for their responses. We are also grateful to the anonymous reviewers for their helpful comments. Special thanks to Saadia Gabriel, Jocelyn Shen, Ashutosh Baheti, and the members of the CMU LTI COMEDY group for their feedback, and OpenAI for providing access to the GPT-3.5 API. This research was supported in part by the Meta Fundamental AI Research Laboratories (FAIR) "*Dynabench Data Collection and Benchmarking Platform*" award "*ContExTox: Context-Aware and Explainable Toxicity Detection.*"

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

## A  Tasks and Datasets

**Formality Data**  We obtain a conversational dataset from Reddit[12] by sampling conversational threads from subreddits such as r/news, r/askscience, and r/Economics (formal conversations), as well as r/movies, r/fantasyfootball, and r/relationships (informal conversations). We focus on two-turn Reddit threads: a parent/preceding context and the response to be rewritten. Next, we sample documents from CNN Daily Mail (formal documents; Nallapati et al., 2016) and the Blog Authorship Corpus (informal documents; Schler et al., 2006). We select four sentences from each data sample: three sentences as the preceding parent context, and the following sentence as the one to be rewritten. For each data sample, we label the context and response using a pre-trained formality classifier.[13]

**Sentiment Data**  We obtain a conversational dataset from the DailyDialog (Li et al., 2017) dataset, focusing on two-turn conversations: a parent/preceding context and the response to be rewritten. Next, we sample entries from the Yelp reviews (Zhang et al., 2015) dataset. Analogous to the document dataset used in formality, we choose four sentences from each data sample: three as the preceding parent context and the subsequent sentence as the one to be rewritten. For each data sample, we annotate the context and response using a sentiment classifier.[14] We partition the data to transform sentences from positive to negative sentiment and vice versa, as well as to convert neutral sentences to positive or negative sentiment.

**Toxicity Data**  We examine three conversational datasets: the Civil Comments in Context (CCC) dataset (Xenos et al., 2021), the Multi-Label Dialogue Malevolence Detection (MDMD) dataset (Zhang et al., 2022), and the ProsocialDialog dataset (Kim et al., 2022). For each dataset, we select two turns from each conversational thread, representing the preceding parent context and the subsequent response as the sentence to be rewritten. We use toxicity scores from PerspectiveAPI,[15]

HateBert[16] and HateRoberta[17] to measure the toxicity of the context and responses.

## B  Modeling Context in Rewriting

We perform few-shot prompting experiments with GPT-3.5 and GPT-NeoX. For GPT-3.5, we use 2 few-shot examples, while for GPT-NeoX, we use 10 few-shot examples. Each few-shot example was manually constructed with the preceding context, an original sentence to be rewritten, an instruction specifying the required style, and a sample rewrite. Figures 6 and 7 display the few-shot prompt examples that we utilized for generating rewrites in the formality change task.

### B.1  In-context learning sample Rewrites

Table 6 shows some additional example rewrites from GPT-3.5 and GPT-NeoX for all tasks.

## C  Contextual Human Evaluation

**Worker selection**  We involve annotators from USA and Canada on Amazon Mechanical Turk (MTurk), who voluntarily opt-in for each task. We recruit annotators for each style transfer task via a corresponding qualification task. In the qualification task, annotators must answer two questions per pair of rewrites: which rewrite has the strongest style strength (e.g., most formal), and which rewrite is the most natural given the preceding context. Annotators assess three pairs of handcrafted rewrites in each qualification task. Those who accurately answer at least five of the six questions (three for style and at least two for naturalness) are approved for the main task. Once approved, we pay them $0.27 USD per head-to-head comparison.

### C.1  Human Evaluation Results

We present the agreement results of the human evaluation studies of detoxification (Table 7), formality change (Table 8) and sentiment transfer (Table 9). Additionally, refer to Figures 4 and 5 for screenshots of the human evaluation instructions provided to annotators and the actual task, respectively.

| inter-rater agreement | StyleStrength | EventMeaning | IntendedMeaning | Naturalness | Fit |
|---|---|---|---|---|---|
| Krippendorff's $\alpha$ | 0.2757 | 0.3778 | 0.4346 | 0.2407 | 0.6855 |
| Fleiss' $\kappa$ | 0.1926 | 0.2906 | 0.3003 | 0.1907 | 0.5167 |

Table 7: Inter-rater agreement scores for human evaluation results of de-toxification task

---

[12]We use `reddit-corpus-small` from http://convokit.cornell.edu/documentation/subreddit.html

[13]https://huggingface.co/s-nlp/roberta-base-formality-ranker

[14]https://huggingface.co/cardiffnlp/twitter-xlm-roberta-base-sentiment

[15]https://perspectiveapi.com/

---

[16]https://huggingface.co/tomh/toxigen_hatebert

[17]https://huggingface.co/tomh/toxigen_roberta

| inter-rater agreement | StyleStrength | EventMeaning | IntendedMeaning | Naturalness | Fit |
|---|---|---|---|---|---|
| Krippendorff's $\alpha$ | 0.6825 | 0.3311 | 0.428 | 0.3551 | 0.4322 |
| Fleiss' $\kappa$ | 0.552 | 0.2504 | 0.2667 | 0.253 | 0.3627 |

Table 8: Inter-rater agreement scores for human evaluation results of formality transfer task

| inter-rater agreement | StyleStrength | EventMeaning | IntendedMeaning | Naturalness | Fit |
|---|---|---|---|---|---|
| Krippendorff's $\alpha$ | 0.1868 | 0.2636 | 0.4292 | 0.3729 | 0.4581 |
| Fleiss' $\kappa$ | 0.121 | 0.1964 | 0.3148 | 0.3581 | 0.2434 |

Table 9: Inter-rater agreement scores for human evaluation results of sentiment change task

# D    Non-contextual Automatic Evaluation

We present the non-contextual automated evaluation results for each task-specific dataset. Figures 16 and 17 illustrate the formality change results for document-level and conversation-level datasets, respectively. Figures 18 and 19 display the sentiment transfer results for document-level and conversation-level datasets, respectively. Figures 27, 21, and 22 depict the de-toxification results for conversational datasets. Notably, all of these figures exhibit similar trends to the aggregate results across all tasks and datasets presented in Figure 3.

## D.1    Correlation with Human Judgments

Effective evaluation metrics should yield judgments that correlate highly with human judgments, assuming that human evaluators represent a gold-standard. For the human judgments along the dimensions of naturalness and fit, we map human preferences as follows: 'contextual' to 1, 'tie' to 0, and 'non-contextual' to $-1$. For the automatic metrics, we assign a score of 1 if a metric scores the contextual rewrite higher than the non-contextual rewrite, and $-1$ if the metric scores are lower for contextual rewrites.

For a given automatic metric and human judgment dimension, we calculate the Spearman rank $\rho$ correlation and Kendall's $\tau$ for the dataset samples used during the contextual human evaluation §5.1. The correlation scores, ranging from $-1$ to 1, are obtained by comparing the mapped automatic scores with the mapped human judgment scores. Higher values indicate a stronger correlation between the scores obtained using the comparison metric and judgments made by human evaluators. Refer to Tables 10 − 12 for the correlation scores of non-contextual evaluation metrics with human judgments for each task.

|  | Lexical ($\rho$) | Semantic ($\rho$) | Lexical ($\tau$) | Semantic ($\tau$) |
|---|---|---|---|---|
| Fit | -0.02 | 0.14 | -0.02 | 0.14 |
| Naturalness | -0.03 | 0.18 | -0.03 | 0.17 |

Table 10: Detoxification task: Spearman rank and Kendall Correlation of non-contextual evaluation metrics with human judgment

|  | Lexical ($\rho$) | Semantic ($\rho$) | Lexical ($\tau$) | Semantic ($\tau$) |
|---|---|---|---|---|
| Fit | 0.18 | 0.28 | 0.17 | 0.27 |
| Naturalness | 0.11 | 0.26 | 0.10 | 0.24 |

Table 11: Formality task: Spearman rank and Kendall Correlation of non-contextual evaluation metrics with human judgment

|  | Lexical ($\rho$) | Semantic ($\rho$) | Lexical ($\tau$) | Semantic ($\tau$) |
|---|---|---|---|---|
| Fit | 0.11 | 0.26 | 0.10 | 0.25 |
| Naturalness | -0.05 | 0.13 | -0.05 | 0.12 |

Table 12: Sentiment task: Spearman rank and Kendall Correlation of non-contextual evaluation metrics with human judgment

|  | Lexical ($\rho$) | Semantic ($\rho$) | CtxSimFit ($\rho$) | Lexical ($\tau$) | Semantic ($\tau$) | CtxSimFit ($\tau$) |
|---|---|---|---|---|---|---|
| Fit | 0.63 | 0.56 | 0.85 | 0.61 | 0.54 | 0.82 |
| Naturalness | 0.59 | 0.58 | 0.88 | 0.56 | 0.55 | 0.84 |

Table 13: Detoxification task: Spearman rank and Kendall Correlation of contextual evaluation metrics with human judgment

|  | Lexical ($\rho$) | Semantic ($\rho$) | CtxSimFit ($\rho$) | Lexical ($\tau$) | Semantic ($\tau$) | CtxSimFit ($\tau$) |
|---|---|---|---|---|---|---|
| Fit | 0.74 | 0.68 | 0.93 | 0.71 | 0.65 | 0.89 |
| Naturalness | 0.68 | 0.69 | 0.94 | 0.65 | 0.66 | 0.90 |

Table 14: Formality task: Spearman rank and Kendall Correlation of contextual evaluation metrics with human judgment

|  | Lexical ($\rho$) | Semantic ($\rho$) | CtxSimFit ($\rho$) | Lexical ($\tau$) | Semantic ($\tau$) | CtxSimFit ($\tau$) |
|---|---|---|---|---|---|---|
| Fit | 0.45 | 0.45 | 0.78 | 0.42 | 0.52 | 0.74 |
| Naturalness | 0.44 | 0.51 | 0.73 | 0.42 | 0.48 | 0.69 |

Table 15: Sentiment task: Spearman rank and Kendall Correlation of contextual evaluation metrics with human judgment

# E    Contextual Automatic Evaluation

We present the contextual automated evaluation results for each task-specific dataset. Figures 23 and 24 illustrate the formality change results for document-level and conversation-level datasets, respectively. Figures 25 and 26 display the sentiment transfer results for document-level and

conversation-level datasets, respectively. Figures 27, 28, and 29 depict the de-toxification results for conversational datasets. All of these figures exhibit similar trends to the aggregate results across all tasks and datasets presented in Figure 3 and they align with the findings from our contextual human evaluation study.

### E.1 Correlation with Human Judgments

Similar to §D.1, we measure the Spearman rank $\rho$ correlation and Kendall's $\tau$ correlation for the samples used during human evaluation in §5.1. Refer to Tables 13 – 15 for the correlation scores of non-contextual evaluation metrics with human judgments for each task.

## Full Instructions    (Expand/Collapse)

### Instructions

Thanks for participating in this new task! We are trying to test out a few automatic systems for rewriting sentences to have a desired formality (formal or informal).

Given a sentence from a conversation or a social media post, a desired or target formality rewriting level (e.g, formal to informal or vice versa) and two rewritten sentences, please answer the following four questions about the rewrites:

- What is the *formality level?*
  Which of the rewrites best exhibits the *required formality (formal or informal)*?

  For example, a sentence which sounds serious/less trivial, polite/less casual might be percieved as *formal* language

  *Formal language* is usually written with correct grammar and vocabulary, in passive voice often with no abbreviations, exclamation marks and imperatives.
  *Informal language* on the other hand is usually with no formal grammar and vocabulary, in active voice with occasional use of abbreviations, exclamation marks and imperatives.

- What are the *described events*?
  Which do you think is *closer in meaning with respect to events in the original* utterance (regardless of the formality change)?

  Who does what to whom?
  Two sentences are similar in event-level details if they describe *roughly the same people, objects and relationships*, irrespective of formality-style attribute differences. But, it's okay if a few details in the sentence get changed a bit.

  For example, when rewriting a news article, it is important to maintain the accuracy of the who, what, when, where, and why aspects of the story.

- What is **intended meaning similarity**?

  Two sentences are similar in their intended meaning if they capture *roughly the same overall message or purpose*. Intended meaning goes beyond the specific details and focuses on *maintaining the author's intention, tone, and the implied meaning* in the rewritten text. But again, it's okay if a few details in the sentence get changed a bit.

  For example, consider an original informal utterance - "Hey, wanna grab some grub later?". When this utterance is rewritten in a more formal style, while maintaining its intended meaning of asking someone to join for a meal, it might be rewritten as "Hello, would you like to join me for a meal later?"

- **Well-suitedness / natural fit**
  Which of the rewritten utterances is *most well suited* to the entire context?

  A rewrite can have the required formality and be similar in meaning to overall content of the original sentence, but it may not necessarily be *fitting / appropriate / relevant / well-suited* in the given context of the dialog or document. The rewrite should be meaningful and coherent while maintaining the flow of the conversation.

### Examples

| Context & Original Sentence | Instruction | Rewrites | Target Formality | Event Similarity | Intended Meaning | Well-suited |
|---|---|---|---|---|---|---|
| *Previous Comment:* This is so crazy. We have people all over this world. cameras everywhere and yet a giant ship manages to go missing for almost ten years. 

 *Response:* Not as many people with cameras in the ocean, duh...And ocean is really big, like REALLY big ;) | Rewrite the response from **informal** to **formal** language. | Fewer people are utilizing cameras in the ocean, as the ocean is immensely vast. | | ✔ | | |
| | | Not as many people have access to cameras in the ocean, of course. Furthermore, the ocean is vast, making it incredibly easy for a vessel to remain undiscovered for nearly a decade. | ✔ | | ✔ | ✔ |
| *Previous Comment:* While reading todays news, I found an interesting text about the investigation on the ISAF and U.S guy, General John Allen. 

 *Response:* The general is now in the FBI's spotlight after discovering lots of mails between him and Jill Kelley. | Rewrite the response from **informal** to **formal** language. | The FBI has initiated an investigation into General John Allen in light of the numerous emails exchanged between him and Jill Kelley. | ✔ | ✔ | | ✔ |
| | | The general is now under the FBI's scrutiny following the revelation of numerous correspondences between him and Jill Kelley. | | | ✔ | |
| *Previous Sentence in document:* Zimbabwe is divided into 8 provinces and 2 cities with provincial status (Harare and Bulawayo). 

 *Response:* The current Legal Aid Directorate (LAD) is staffed by 15 lawyers, all based in Harare, representing the needs of Zimbabwe's 12 million citizens. | Rewrite the response from **formal** to **informal** language. | LAD's got 15 lawyers based in Harare, all working to meet the needs of Zimbabwe's 12 million people. | ✔ | | | |
| | | The Legal Aid Directorate has 15 lawyers in Harare who take care of the needs of the 12 million people living in Zimbabwe. | | ✔ | ✔ | ✔ |

Figure 4: Screenshot of the instructions for human evaluation annotation

## Task

${context_type}:
${context}

${original_type} (original):
${original}

**Instruction:** Rewrite the original sentence/response from ${style1} to **${style2}** language.

## Rewrites:

rewrite A:
${rewriteA}

rewrite B:
${rewriteB}

Q1: Pretend like you were the person that said the original sentence, which would you prefer as a ${style2} rewrite?
Pick the rewrite which you'd pick as more natural to you. Do not over think this, pick which you'd like as rewrite if you were the person who said the original sentence.

○ Rewrite A                    ${rewriteA}

○ Rewrite B                    ${rewriteB}

○ Either Option                Both rewrites could work / Cannot decide

Q2: How confident do you feel about your preferred choice of rewrite?
Pick your confidence level

| 10% | 20% | 30% | 40% | 50% | 60% | 70% | 80% | 90% | 100% |

Q3: Which of these rewrites is **most ${style2}**?
Pick the rewrite which is most likely to be associated most ${style2} language regardless of how much the meaning of the original sentence has changed.

○ Rewrite A                    ${rewriteA}

○ Rewrite B                    ${rewriteB}

○ Either Option                Both rewrites could work / Cannot decide

Q4: Which rewrite do you think is **closer to the described events** in the original sentence?
Choose the sentence with the same people, things, and relationships/connections as the original sentence.

○ Rewrite A                    ${rewriteA}

○ Rewrite B                    ${rewriteB}

○ Either Option                Both rewrites could work / Cannot decide

Q5: Which rewrite do you think is **closer to the intended meaning**?
Pick the sentence that has the closest implied overall message.

○ Rewrite A                    ${rewriteA}

○ Rewrite B                    ${rewriteB}

○ Either Option                Both rewrites could work / Cannot decide

Q6: Which sentence is best **well-suited / relevant / flows better / specific** with the overall preceding conversation / document flow?
Try to pick the rewrite that is most coherent and fitting with the preceding context.

○ Rewrite A                    ${rewriteA}

○ Rewrite B                    ${rewriteB}

○ Either Option                Both rewrites could work / Cannot decide

Q7. On a scale of 1-5, how much should the original sentence should be changed to really rewrite it to be ${style2}?

All of it needs                                                        Barely anything
to change

| 1 | 2 | 3 | 4 | 5 |

Figure 5: Screenshot of the human evaluation annotation task

| Task | Conversion | Context | Original Sentence | Model | Contextual Rewrite | Non-contextual Rewrite |
|------|-----------|---------|-------------------|-------|--------------------|------------------------|
| (a) Formality | Informal -> Formal | 📄 I am reading a book, 777 Mathematical Conversation Starters by John dePillis. Its an interesting book with lots of quotes, poems | ❯❯ Herez some of them, and further I will keep updating this page with what I like. | GPT-3.5 | ❯❯ I plan to continue to update this page with further selections that I find particularly intriguing from the book. | ❯❯ Here are some of them, and I will continue to update this page with my preferences. |
| | | | | GPT-NeoX | ❯❯ Here are some of them, and in a later post I will keep updating this page with my favourite quotes from the book. | ❯❯ Here is some of the one and further I will update this page with what else I find interesting. |
| (b) Formality | Semi-formal -> Informal | 💬 I can never again take a site serious that gives the mess and ... just an awful game. | 💬 I liked that game. It's almost as if opinions are subjective. | GPT-3.5 | 💬 I enjoyed Ni No Kuni 2, but it's clear that other people didn't - and that's okay. Everyone's got their own opinion! | 💬 I enjoyed that game. It seems opinions may indeed vary among individuals. |
| | | | | GPT-NeoX | 💬 I liked that game, despite critical opinions to the contrary. The fact that people enjoyed it doesn't make it any worse than anything else. | 💬 I liked that game. It's almost as if opinions are relative. |
| (c) Detoxification | Toxic -> Less Toxic | 💬 Did Rolando just call me just to tell me shut up? | 💬 @USER fuck yeah I did!! Shut up | GPT-3.5 | 💬 @USER Yes, I was asking you to be quiet. | 💬 I am very proud of my accomplishment! yeah |
| | | | | GPT-NeoX | 💬 user yes, please keep quiet | 💬 yeah, I did! Shut up. |
| (d) Detoxification | Toxic -> Less Toxic | 💬 "i am here 2 play games i am playin 1 wit u 10:50 to 2:42 baby i'm everywhare" | 💬 It's so amusing when the illiterate try to taunt. | GPT-3.5 | 💬 I'm not sure I understand what you're trying to say. Could you please rephrase? | 💬 It's interesting to see how someone is illiterate tries to tease. |
| | | | | GPT-NeoX | 💬 It's amusing when someone attempts to insult or taunt me and they are clearly uneducated about language. | 💬 It's funny when the uneducated try to insult. |
| (e) Sentiment | Positive -> Negative | 📄 I just want to echo what everyone else has already said about this mechanic... Not only did Tim fix my car quickly and inexpensively, he actually came to pick it up himself so that I wouldn't have to spend money on a tow truck! | ❯❯ He's also an extremely friendly, funny guy who puts you at ease immediately. | GPT-3.5 | ❯❯ Despite being an efficient mechanic, Tim lacked the friendly and humorous demeanor expected of a service provider. | ❯❯ He's an unfriendly and humorless person who makes you feel uneasy right away. |
| | | | | GPT-NeoX | ❯❯ However, Tim's not very friendly or humorous which might make you feel uneasy | ❯❯ He is also a very rude, unamused guy who makes you feel anxious. |
| (f) Sentiment | Negative -> Positive | 💬 I had to move it so that I can get my bike through. | 💬 How dare you be so inconsiderate ? That's wrong. | GPT-3.5 | 💬 It was very considerate of you to move it, so that you can get your bike through. That's great. | 💬 How kind of you to be so considerate! That's right. |
| | | | | GPT-NeoX | 💬 I understand you needed to move it so you could get your bike through - thank you for being thoughtful! | 💬 How can you be so considerate! That's right. |

Table 6: Examples from our datasets and corresponding rewrites generated by GPT-3.5 and GPT-NeoX, showcasing all three tasks: formality change, de-toxification, and sentiment transfer. 📄 = document context, ❯❯ = sentence to rewrite, ❯❯ = contextual rewrite, ❯❯ = non-contextual rewrite; 💬 = previous turn in conversation, 💬 = response to rewrite, 💬 = contextual rewrite of response, 💬 = non-contextual rewrite of response

| model | rewrite type | *Lexical* | | | *Semantic* | | | *Fluency* | *Style* |
|---|---|---|---|---|---|---|---|---|---|
| | | ROU | MET | WER | BERT-S | SBERT | Smatch | PPL | |
| GPT-3.5 | contextual | 0.19 | 0.40 | 2.14 | 0.92 | 0.62 | 0.51 | 38.37 | 0.59 |
| | non-contextual | 0.28 | 0.49 | 0.91 | 0.94 | 0.73 | 0.67 | 43.40 | 0.58 |
| | random-context | 0.18 | 0.35 | 3.17 | 0.91 | 0.52 | 0.47 | | |
| GPT-NeoX | contextual | 0.26 | 0.42 | 1.88 | 0.91 | 0.60 | 0.47 | 44.59 | 0.42 |
| | non-contextual | 0.45 | 0.63 | 0.72 | 0.95 | 0.80 | 0.67 | 44.80 | 0.35 |
| | random-context | 0.21 | 0.36 | 1.90 | 0.91 | 0.60 | 0.41 | | |

Table 16: Non-contextual Automatic Evaluation Results on **Formality**: Document-level context from CNN/DailyMail + Blog Authorship Corpus

| model | rewrite type | *Lexical* | | | *Semantic* | | | *Fluency* | *Style* |
|---|---|---|---|---|---|---|---|---|---|
| | | ROU | MET | WER | BERT-S | SBERT | Smatch | PPL | |
| GPT-3.5 | contextual | 0.16 | 0.38 | 2.67 | 0.90 | 0.67 | 0.45 | 33.78 | 0.68 |
| | non-contextual | 0.22 | 0.41 | 1.23 | 0.91 | 0.72 | 0.53 | 40.06 | 0.67 |
| | random-context | 0.15 | 0.32 | 3.72 | 0.89 | 0.58 | 0.43 | | |
| GPT-NeoX | contextual | 0.24 | 0.41 | 1.97 | 0.90 | 0.65 | 0.44 | 52.45 | 0.45 |
| | non-contextual | 0.36 | 0.55 | 0.98 | 0.92 | 0.78 | 0.54 | 57.12 | 0.37 |
| | random-context | 0.27 | 0.40 | 2.70 | 0.90 | 0.60 | 0.44 | | |

Table 17: Non-contextual Automatic Evaluation Results on **Formality**: Conversational context comprised of Reddit threads

| model | rewrite type | *Lexical* | | | *Semantic* | | | *Fluency* | *Style* |
|---|---|---|---|---|---|---|---|---|---|
| | | ROU | MET | WER | BERT-S | SBERT | Smatch | PPL | |
| GPT-3.5 | contextual | 0.18 | 0.36 | 1.57 | 0.90 | 0.59 | 0.40 | 42.21 | 0.74 |
| | non-contextual | 0.40 | 0.61 | 0.64 | 0.94 | 0.80 | 0.63 | 58.38 | 0.64 |
| | random-context | 0.14 | 0.30 | 1.74 | 0.89 | 0.49 | 0.36 | | |
| GPT-NeoX | contextual | 0.27 | 0.43 | 1.43 | 0.91 | 0.56 | 0.41 | 57.64 | 0.49 |
| | non-contextual | 0.45 | 0.60 | 0.67 | 0.94 | 0.74 | 0.62 | 73.02 | 0.49 |
| | random-context | 0.29 | 0.44 | 1.37 | 0.91 | 0.56 | 0.42 | | |

Table 18: Non-contextual Automatic Evaluation Results on **Sentiment**: Document-level context comprised of Yelp Reviews

| model | rewrite type | *Lexical* | | | *Semantic* | | | *Fluency* | *Style* |
|---|---|---|---|---|---|---|---|---|---|
| | | ROU | MET | WER | BERT-S | SBERT | Smatch | PPL | |
| GPT-3.5 | contextual | 0.30 | 0.54 | 1.03 | 0.91 | 0.63 | 0.53 | 36.31 | 0.69 |
| | non-contextual | 0.45 | 0.67 | 0.65 | 0.93 | 0.75 | 0.68 | 42.82 | 0.64 |
| | random-context | 0.30 | 0.52 | 1.07 | 0.91 | 0.59 | 0.53 | | |
| GPT-NeoX | contextual | 0.16 | 0.30 | 1.66 | 0.87 | 0.43 | 0.22 | 42.39 | 0.35 |
| | non-contextual | 0.33 | 0.48 | 0.81 | 0.90 | 0.59 | 0.50 | 64.09 | 0.25 |
| | random-context | 0.18 | 0.33 | 1.63 | 0.88 | 0.43 | 0.30 | | |

Table 19: Non-contextual Automatic Evaluation Results on **Sentiment**: Conversational context from DailyDialog dataset

| model | rewrite type | *Lexical* | | | *Semantic* | | | *Fluency* | *Style* | *Style* | *Style* |
|---|---|---|---|---|---|---|---|---|---|---|---|
| | | ROU | MET | WER | BERT-S | SBERT | Smatch | PPL | *HateRoberta* | *HateBert* | *Perspective* |
| GPT-3.5 | contextual | 0.20 | 0.36 | 0.94 | 0.90 | 0.64 | 0.45 | 37.92 | 0.01 | 0.41 | 0.06 |
| | non-contextual | 0.24 | 0.41 | 0.80 | 0.91 | 0.72 | 0.51 | 40.98 | 0.01 | 0.47 | 0.07 |
| | random-context | 0.17 | 0.32 | 0.97 | 0.89 | 0.57 | 0.43 | | | | |
| GPT-NeoX | contextual | 0.32 | 0.40 | 0.78 | 0.90 | 0.60 | 0.46 | 63.04 | 0.07 | 0.61 | 0.13 |
| | non-contextual | 0.44 | 0.52 | 0.61 | 0.92 | 0.71 | 0.57 | 67.47 | 0.10 | 0.69 | 0.15 |
| | random-context | 0.32 | 0.40 | 0.77 | 0.90 | 0.58 | 0.47 | | | | |

Table 20: Non-contextual Automatic Evaluation Results on **Toxicity**: Conversational context from CCC dataset

| model | rewrite type | Lexical | | | Semantic | | | Fluency | Style | Style | Style |
|---|---|---|---|---|---|---|---|---|---|---|---|
| | | ROU | MET | WER | BERT-S | SBERT | Smatch | PPL | HateRoberta | HateBert | Perspective |
| GPT-3.5 | contextual | 0.11 | 0.32 | 1.18 | 0.87 | 0.51 | 0.43 | 75.48 | 0.04 | 0.31 | 0.11 |
| | non-contextual | 0.12 | 0.34 | 0.99 | 0.88 | 0.56 | 0.47 | 78.23 | 0.05 | 0.34 | 0.12 |
| | random-context | 0.08 | 0.28 | 1.24 | 0.86 | 0.42 | 0.40 | | | | |
| GPT-NeoX | contextual | 0.18 | 0.28 | 1.29 | 0.87 | 0.45 | 0.39 | 80.10 | 0.39 | 0.62 | 0.35 |
| | non-contextual | 0.32 | 0.49 | 0.91 | 0.90 | 0.67 | 0.54 | 106.66 | 0.52 | 0.74 | 0.46 |
| | random-context | 0.15 | 0.25 | 1.41 | 0.86 | 0.39 | 0.36 | | | | |

Table 21: Non-contextual Automatic Evaluation Results on **Toxicity**: Conversational context from MDMD dataset

| model | rewrite type | Lexical | | | Semantic | | | Fluency | Style | Style | Style |
|---|---|---|---|---|---|---|---|---|---|---|---|
| | | ROU | MET | WER | BERT-S | SBERT | Smatch | PPL | HateRoberta | HateBert | Perspective |
| GPT-3.5 | contextual | 0.05 | 0.21 | 1.69 | 0.88 | 0.38 | 0.29 | 22.80 | 0.03 | 0.25 | 0.06 |
| | non-contextual | 0.11 | 0.29 | 0.97 | 0.91 | 0.52 | 0.41 | 33.00 | 0.14 | 0.40 | 0.09 |
| | random-context | 0.05 | 0.19 | 1.61 | 0.88 | 0.25 | 0.29 | | | | |
| GPT-NeoX | contextual | 0.25 | 0.40 | 1.12 | 0.91 | 0.53 | 0.44 | 32.86 | 0.37 | 0.63 | 0.26 |
| | non-contextual | 0.43 | 0.59 | 0.66 | 0.94 | 0.72 | 0.63 | 37.90 | 0.64 | 0.79 | 0.38 |
| | random-context | 0.25 | 0.40 | 1.06 | 0.91 | 0.48 | 0.44 | | | | |

Table 22: Non-contextual Automatic Evaluation Results on **Toxicity**: Conversational context from ProsocialDialog dataset

| model | rewrite type | Lexical | | | Semantic | | | Coherence | Cohesiveness | Custom |
|---|---|---|---|---|---|---|---|---|---|---|
| | | ROU$^{ctx}$ | MET$^{ctx}$ | WER$^{ctx}$ | BERT-S$^{ctx}$ | SBERT$^{ctx}$ | Smatch$^{ctx}$ | PPL$^{ctx}$ | NSP | CtxSimFit |
| GPT-3.5 | contextual | 0.16 | 0.23 | 0.89 | 0.90 | 0.61 | 0.38 | 22.05 | 0.94 | 0.93 |
| | non-contextual | 0.15 | 0.20 | 0.87 | 0.89 | 0.50 | 0.33 | 32.79 | 0.87 | 0.91 |
| | random-context | 0.10 | 0.16 | 1.06 | 0.87 | 0.38 | 0.29 | 34.24 | 0.69 | 0.80 |
| GPT-NeoX | contextual | 0.25 | 0.28 | 0.83 | 0.90 | 0.62 | 0.37 | 20.51 | 0.97 | 0.94 |
| | non-contextual | 0.23 | 0.24 | 0.84 | 0.89 | 0.52 | 0.33 | 32.15 | 0.94 | 0.94 |
| | random-context | 0.16 | 0.19 | 0.92 | 0.87 | 0.42 | 0.29 | 39.18 | 0.82 | 0.87 |

Table 23: Contextual Automatic Evaluation Results on **Formality**: Document-level context from CNN/DailyMail + Blog Authorship Corpus

| model | rewrite type | Lexical | | | Semantic | | | Coherence | Cohesiveness | Custom |
|---|---|---|---|---|---|---|---|---|---|---|
| | | ROU$^{ctx}$ | MET$^{ctx}$ | WER$^{ctx}$ | BERT-S$^{ctx}$ | SBERT$^{ctx}$ | Smatch$^{ctx}$ | PPL$^{ctx}$ | NSP | CtxSimFit |
| GPT-3.5 | contextual | 0.14 | 0.27 | 0.91 | 0.89 | 0.66 | 0.37 | 28.82 | 0.88 | 0.89 |
| | non-contextual | 0.14 | 0.24 | 0.95 | 0.88 | 0.56 | 0.36 | 41.02 | 0.82 | 0.87 |
| | random-context | 0.11 | 0.20 | 1.58 | 0.87 | 0.47 | 0.32 | 46.79 | 0.73 | 0.81 |
| GPT-NeoX | contextual | 0.21 | 0.29 | 0.88 | 0.89 | 0.64 | 0.36 | 34.74 | 0.90 | 0.90 |
| | non-contextual | 0.23 | 0.29 | 0.88 | 0.88 | 0.58 | 0.36 | 52.45 | 0.86 | 0.89 |
| | random-context | 0.17 | 0.22 | 1.29 | 0.87 | 0.46 | 0.31 | 53.98 | 0.80 | 0.85 |

Table 24: Contextual Automatic Evaluation Results on **Formality**: Conversational context comprised of Reddit threads

| model | rewrite type | Lexical | | | Semantic | | | Coherence | Cohesiveness | Custom |
|---|---|---|---|---|---|---|---|---|---|---|
| | | ROU$^{ctx}$ | MET$^{ctx}$ | WER$^{ctx}$ | BERT-S$^{ctx}$ | SBERT$^{ctx}$ | Smatch$^{ctx}$ | PPL$^{ctx}$ | NSP | CtxSimFit |
| GPT-3.5 | contextual | 0.11 | 0.17 | 0.92 | 0.88 | 0.53 | 0.24 | 25.53 | 0.98 | 0.94 |
| | non-contextual | 0.16 | 0.17 | 0.87 | 0.88 | 0.48 | 0.24 | 41.18 | 0.93 | 0.94 |
| | random-context | 0.07 | 0.12 | 0.93 | 0.86 | 0.38 | 0.19 | 44.13 | 0.82 | 0.86 |
| GPT-NeoX | contextual | 0.13 | 0.16 | 0.90 | 0.87 | 0.47 | 0.23 | 33.05 | 0.96 | 0.93 |
| | non-contextual | 0.17 | 0.18 | 0.87 | 0.88 | 0.44 | 0.22 | 48.59 | 0.93 | 0.93 |
| | random-context | 0.12 | 0.15 | 0.91 | 0.87 | 0.41 | 0.20 | 53.44 | 0.91 | 0.91 |

Table 25: Contextual Automatic Evaluation Results on **Sentiment**: Document-level context comprised of Yelp Reviews

| model | rewrite type | Lexical | | | Semantic | | | Coherence | Cohesiveness | Custom |
|---|---|---|---|---|---|---|---|---|---|---|
| | | ROU[ctx] | MET[ctx] | WER[ctx] | BERT-S[ctx] | SBERT[ctx] | Smatch[ctx] | PPL[ctx] | NSP | CtxSimFit |
| GPT-3.5 | contextual | 0.25 | 0.36 | 0.84 | 0.89 | 0.62 | 0.43 | 33.88 | 0.97 | 0.94 |
| | non-contextual | 0.28 | 0.35 | 0.81 | 0.89 | 0.54 | 0.41 | 50.60 | 0.92 | 0.93 |
| | random-context | 0.20 | 0.30 | 0.89 | 0.88 | 0.45 | 0.38 | 54.10 | 0.87 | 0.89 |
| GPT-NeoX | contextual | 0.17 | 0.26 | 0.97 | 0.86 | 0.46 | 0.34 | 32.45 | 0.88 | 0.88 |
| | non-contextual | 0.20 | 0.24 | 0.87 | 0.87 | 0.42 | 0.21 | 60.31 | 0.86 | 0.88 |
| | random-context | 0.12 | 0.21 | 1.04 | 0.85 | 0.34 | 0.21 | 41.69 | 0.79 | 0.83 |

Table 26: Contextual Automatic Evaluation Results on **Sentiment**: Conversational context from DailyDialog dataset

| model | rewrite type | Lexical | | | Semantic | | | Coherence | Cohesiveness | Custom |
|---|---|---|---|---|---|---|---|---|---|---|
| | | ROU[ctx] | MET[ctx] | WER[ctx] | BERT-S[ctx] | SBERT[ctx] | Smatch[ctx] | PPL[ctx] | NSP | CtxSimFit |
| GPT-3.5 | contextual | 0.16 | 0.24 | 0.86 | 0.88 | 0.61 | 0.36 | 28.45 | 0.95 | 0.93 |
| | non-contextual | 0.17 | 0.25 | 0.85 | 0.88 | 0.57 | 0.35 | 37.40 | 0.91 | 0.91 |
| | random-context | 0.12 | 0.20 | 0.90 | 0.87 | 0.47 | 0.31 | 38.96 | 0.89 | 0.89 |
| GPT-NeoX | contextual | 0.24 | 0.25 | 0.82 | 0.88 | 0.54 | 0.34 | 37.41 | 0.96 | 0.93 |
| | non-contextual | 0.29 | 0.29 | 0.77 | 0.89 | 0.54 | 0.32 | 51.56 | 0.92 | 0.92 |
| | random-context | 0.21 | 0.23 | 0.84 | 0.87 | 0.44 | 0.32 | 52.24 | 0.89 | 0.90 |

Table 27: Contextual Automatic Evaluation Results on **Toxicity**: Conversational context from CCC dataset

| model | rewrite type | Lexical | | | Semantic | | | Coherence | Cohesiveness | Custom |
|---|---|---|---|---|---|---|---|---|---|---|
| | | ROU[ctx] | MET[ctx] | WER[ctx] | BERT-S[ctx] | SBERT[ctx] | Smatch[ctx] | PPL[ctx] | NSP | CtxSimFit |
| GPT-3.5 | contextual | 0.10 | 0.22 | 0.91 | 0.86 | 0.50 | 0.34 | 49.50 | 0.96 | 0.92 |
| | non-contextual | 0.08 | 0.19 | 0.92 | 0.86 | 0.40 | 0.31 | 70.91 | 0.86 | 0.87 |
| | random-context | 0.06 | 0.16 | 0.96 | 0.84 | 0.29 | 0.29 | 71.09 | 0.82 | 0.84 |
| GPT-NeoX | contextual | 0.20 | 0.25 | 0.85 | 0.87 | 0.52 | 0.35 | 40.79 | 0.93 | 0.90 |
| | non-contextual | 0.19 | 0.26 | 0.87 | 0.87 | 0.47 | 0.36 | 90.64 | 0.89 | 0.90 |
| | random-context | 0.09 | 0.15 | 0.97 | 0.85 | 0.30 | 0.26 | 92.21 | 0.76 | 0.81 |

Table 28: Contextual Automatic Evaluation Results on **Toxicity**: Conversational context from MDMD dataset

| model | rewrite type | Lexical | | | Semantic | | | Coherence | Cohesiveness | Custom |
|---|---|---|---|---|---|---|---|---|---|---|
| | | ROU[ctx] | MET[ctx] | WER[ctx] | BERT-S[ctx] | SBERT[ctx] | Smatch[ctx] | PPL[ctx] | NSP | CtxSimFit |
| GPT-3.5 | contextual | 0.09 | 0.18 | 0.90 | 0.89 | 0.54 | 0.29 | 14.89 | 0.98 | 0.93 |
| | non-contextual | 0.06 | 0.12 | 0.93 | 0.88 | 0.36 | 0.21 | 29.75 | 0.89 | 0.90 |
| | random-context | 0.03 | 0.11 | 0.95 | 0.86 | 0.21 | 0.22 | 32.58 | 0.84 | 0.86 |
| GPT-NeoX | contextual | 0.19 | 0.23 | 0.85 | 0.89 | 0.52 | 0.32 | 18.98 | 0.94 | 0.93 |
| | non-contextual | 0.20 | 0.22 | 0.85 | 0.89 | 0.44 | 0.30 | 34.42 | 0.88 | 0.91 |
| | random-context | 0.12 | 0.17 | 0.90 | 0.88 | 0.33 | 0.26 | 36.32 | 0.84 | 0.88 |

Table 29: Contextual Automatic Evaluation Results on **Toxicity**: Conversational context from ProsocialDialog dataset

Here is the preceding context for some text: {While reading todays news, I found an interesting text about the investigation on the ISAF and U.S guy, General John Allen. }
Here is the text: {The general is now in the FBI's spotlight after discovering lots of mails between him and Jill Kelley. }
Here is a rewrite of the text from informal to formal langauge, under the given context.
{The FBI has initiated an investigation into General John Allen in light of the numerous emails exchanged between him and Jill Kelley.}
###
Here is the preceding context for some text: {From the fourth policy year onwards, your boat is insured for its current value. This sailing ground covers the waters of the Baltic, including Kattegat and Skagerrak, as well the European rivers and inland waterways.}
Here is the text: {In estuaries, an imaginary continuation of the coastline forms the boundary between the inland waterways and the neighbouring sea.}
Here is a rewrite of the text from formal to informal language, under the given context.
{In the estuaries, a pretend line along the shore marks the line between the canals and the sea.}
###
Here is the preceding context for some text: {YACHTING24 does not pass on any data to third parties. }
Here is the text: {Compliance with data protection regulations is monitored by our in-house data protection officer in your interests.}
Here is a rewrite of the text from formal to informal language, under the given context.
{We don't give out your data to anyone else, and our data protection officer keeps an eye on everything to make sure your info's secure.}
###
Here is the preceding context for some text: {If arrested with a warrant the defendant must be brought to a police station or charging station as soon as possible.}
Here is the text: {A defendant may be detained for no more than 48 hours unless a magistrate authorizes an extension of the detention.}
Here is a rewrite of the text from formal to informal language, under the given context.
{The defendant can't be held for more than two days unless a judge says it's okay.}
###
Here is the preceding context for some text: {Zimbabwe is divided into 8 provinces and 2 cities with provincial status (Harare and Bulawayo).}
Here is the text: {The current Legal Aid Directorate (LAD) is staffed by 15 lawyers, all based in Harare, representing the needs of Zimbabwe's 12 million citizens}
Here is a rewrite of the text from formal to informal language, under the given context.
{The Legal Aid Directorate has 15 lawyers in Harare who take care of the needs of the 12 million people living n Zimbabwe.}
###
Here is the preceding context for some text: {I heard that trans fat is bad for the body. Why is that?}
Here is the text: {Where can we find it in our daily food?}
Here is a rewrite of the text from informal to formal language, under the given context.
{What foods are likely to contain trans fat that we consume on a daily basis?}
###
Here is the preceding context for some text: {It was because the number of centroids I have is actually larger than 1M so it doesn't let me train…however I'm not entirely sure what number of centroids I need either.}
Here is the text: {I'm just directly keeping the ratio of centroids to total vectors constant from earlier code.}
Here is a rewrite of the text from informal to formal language, under the given context.
{I am maintaining a consistent ratio of centroids to total vectors from my prior code, despite the number of centroids exceeding one million.}
###
Here is the preceding context for some text: {In the san francisco bay area, does it make sense to rent or buy ?}
Here is the text: {the prices of rent and the price of buying does not make sense to me, mostly the rent will not cover the mortgage}
Here is a rewrite of the text from informal to formal language, under the given context.
{The costs of renting and purchasing property in the San Francisco Bay Area do not appear to be in balance, as rent payments generally do not cover the mortgage.}
###
Here is the preceding context for some text: {I have very small stuff stuck under my keyboard and it prevents it to from working.}
Here is the text: {What will be the best way to clean it?}
Here is a rewrite of the text from informal to formal language, under the given context.
{What would be the most efficient method of removing the debris that is obstructing the keyboard?}
###
Here is the preceding context for some text: {Blushing is a unique blend of evolutionary and social behavior.}
Here is the text: {It's an involuntary reaction of the sympathetic nervous system, which is responsible for our ""fight or flight"" response, but blushing is solely triggered by social cues.}
Here is a rewrite of the text from informal to formal language, under the given context.
{Blushing is an involuntary physiological response of the sympathetic nervous system, which is typically activated in response to a perceived fear or threat. However, blushing is activated exclusively by social stimuli.}
###

Figure 6: **Formality**: 10-shot prompting examples for GPT-NeoX

Statement: While reading todays news, I found an interesting text about the investigation on the ISAF and U.S guy, General John Allen.
Response: The general is now in the FBI's spotlight after discovering lots of mails between him and Jill Kelley.
Rewrite the Response from informal to formal language, under the given context.
Rewrite: The FBI has initiated an investigation into General John Allen in light of the numerous emails exchanged between him and Jill Kelley.

Statement: Zimbabwe is divided into 8 provinces and 2 cities with provincial status (Harare and Bulawayo).
Response: The current Legal Aid Directorate (LAD) is staffed by 15 lawyers, all based in Harare, representing the needs of Zimbabwe's 12 million citizens
Rewrite the Response from formal to informal language, under the given context.
Rewrite: The Legal Aid Directorate has 15 lawyers in Harare who take care of the needs of the 12 million people living in Zimbabwe.

Figure 7: **Formality**: 2-shot prompting examples for GPT-3.5