# OpenReview forum: "$\textit{``Don't Take This Out of Context!''}$ On the Need for Contextual Models and Evaluations for Stylistic Rewriting"
_EMNLP/2023/Conference — EMNLP 2023 Main_

### Official Review · Reviewer_fCPE · 2023-08-03

**Soundness:** 4

**Excitement:**

4: Strong: This paper deepens the understanding of some phenomenon or lowers the barriers to an existing research direction.

**Missing References:**

NA

**Paper Topic And Main Contributions:**

The paper focuses on the significance of contextual modeling for stylistic re-writing generation and evaluation.


**Questions For The Authors:**

Q1: Among Tables 12-14, there is a lower correlation for the CtxSimFit score in the Sentiment task (Table 14). It would be helpful to explain the reasons behind this observation and discuss any possible implications for future research or improvements.

Q2: Are there a taxonomy of the errors made by (non)contextual models?

**Reasons To Accept:**

Strengths:
- The paper's recognition of the importance of context in both stylistic re-writing generation and evaluation is crucial for the field.
- The evaluation of existing non-contextual and contextual systems, especially against human judgment, is a valuable empirical effort that establishes the state-of-the-art in this area.
- The introduction of CtxSimFit as a new score that correlates better with human judgment appears to be effective.

Based on these merits, I strongly recommend accepting this paper (Excitement=4, Soundness=4).

**Reasons To Reject:**

Weaknesses:
As a non-expert in the field of stylistic re-writing, I can only identify general weaknesses, and upon my initial review, I find no reason to reject this paper.

Challenges:

- Line 206: The authors mention utilizing large language models (LLMs) and in-context learning for the task. It would be beneficial to explain why they specifically chose these approaches and whether their conclusions would still hold for smaller models.
- Line 452: The terms "Coherence" and "Cohesiveness" are formal linguistic concepts that require careful definitions in the given context. Please provide a clear definition here. Additionally, there are concerns about using pplx^Ctx as a proper metric for evaluating text coherence, given the availability of specialized methods for this purpose [1].
- Line 468: What do you mean by NSP head? Since the equation at Line 474 requires a score calculated by NSP(C, X), but NSP head is a compoment in neural network architecture. What's more, there are more clarifications to be made here: (1) Why $\alpha$ is set to be 0.5? Won't other values yield better results? (2) What is the intuition behind this formula? Is it modeling coheisveness or coherence?

[1] Summeval: Re-evaluating summarization evaluation. TACL 2021.

**Reproducibility:**

4: Could mostly reproduce the results, but there may be some variation because of sample variance or minor variations in their interpretation of the protocol or method.

**Reviewer Confidence:**

3: Pretty sure, but there's a chance I missed something. Although I have a good feel for this area in general, I did not carefully check the paper's details, e.g., the math, experimental design, or novelty.

**Typos Grammar Style And Presentation Improvements:**

Tables 12 to 14 are important results that demonstrate CtxSimFit's correlation with human judgment. It would be beneficial to find a way to incorporate these tables into the main text for better visibility and understanding.

---

> ### Author Rebuttal · Authors · 2023-08-29
>
> We thank the reviewer for the thoughtful comments and positive feedback on “the paper's recognition of the importance of context in both stylistic re-writing generation and evaluation is crucial for the field ”.
>
> 1. `Line 206: Choice of LLMs and validity of results for smaller models`\
> We incorporated textual context using few-shot prompting of LLMs, which have shown strong rewriting capabilities with and without context. Existing generative models for chit-chat or goal-oriented dialogue agents struggle to effectively incorporate context. On the other hand, custom-made models often do not consider context. Please see response 3 to Reviewer [XbBa](https://openreview.net/forum?id=In4L79U5n7&noteId=43bvMBdDLS) for additional details on the choice of LLMs.
> \
> \
> As for the validity of the results for smaller models, we believe the results still hold since there is some preliminary research [1] that explored context in small custom-trained seq2seq rewriting models which showed that contextual rewrites are preferred by human annotators.
> \
> \
> [1] Cheng, Yu, Zhe Gan, Yizhe Zhang, Oussama Elachqar, Dianqi Li, and Jingjing Liu. "Contextual Text Style Transfer." In Findings of the Association for Computational Linguistics: EMNLP 2020, pp. 2915-2924. 2020.
> \
> &nbsp;
> 2. `Line 452: Definitions of  "Coherence" and "Cohesiveness"; Clarification on using pplx^Ctx for coherence`\
> The terms coherence and cohesion in linguistics are commonly defined as follows [2]:
> \
> *Cohesion:* sentence pairs fitting together the way two pieces of a jigsaw puzzle do
> \
> *Coherence:* what all the sentences in a piece of writing add up to, the way all the pieces in a puzzle add up to the picture on the box
> \
> \
> However, these concepts have varying definitions and are slightly adapted differently for various tasks. We computed coherence in our study by measuring the perplexity of the rewrite conditioned on the context using GPT-2, based on the above linguistic definitions and prior research [4, 5]. We appreciate the reviewer for mentioning the work of Fabbri, Alexander R., et al [6],  which is specific to summarization and suggests that metrics like ROUGE-4 and character-based n-gram overlap are highly correlated with coherence. In our paper, we included ROUGE-2 under "lexical similarity", and in the final version, we will also add ROUGE-4 to our evaluation suite.
> \
> \
> Similarly following prior work based on these definitions [7], we compute cohesiveness using the probabilities from the next sentence prediction head of BERT which measures if the context followed by the rewrite is fitting.
> \
> \
> We will acknowledge these varying definitions in our paper and clarify the specific meaning we intend to convey.
> \
> \
> [2] J.M. Williams and G.G. Colomb. 1995. Style: Toward Clarity and Grace. Chicago guides to writing, editing, and publishing. University of Chicago Press.\
> [3] See, Abigail, Aneesh Pappu, Rohun Saxena, Akhila Yerukola, and Christopher D. Manning. "Do Massively Pretrained Language Models Make Better Storytellers?." In Proceedings of the 23rd Conference on Computational Natural Language Learning (CoNLL), pp. 843-861. 2019.\
> [4] Pang, Bo, Erik Nijkamp, Wenjuan Han, Linqi Zhou, Yixian Liu, and Kewei Tu. "Towards Holistic and Automatic Evaluation of Open-Domain Dialogue Generation." In Proceedings of the 58th Annual Meeting of the Association for Computational Linguistics, pp. 3619-3629. 2020.\
> [5] Fabbri, Alexander R., Wojciech Kryściński, Bryan McCann, Caiming Xiong, Richard Socher, and Dragomir Radev. "Summeval: Re-evaluating summarization evaluation." Transactions of the Association for Computational Linguistics 9 (2021): 391-409.\
> [6] Sugiyama, Hiroaki. "Dialogue breakdown detection using BERT with traditional dialogue features." In Increasing Naturalness and Flexibility in Spoken Dialogue Interaction: 10th International Workshop on Spoken Dialogue Systems, pp. 419-427. Springer Singapore, 2021.
> \
> &nbsp;
> 3. `Line 468: BERT’s Next Sentence Prediction (NSP) head for Cohesiveness`\
> In addition to the masked language modeling pre-training objective, BERT has an additional “next sentence prediction” (NSP) task that jointly pre-trains text-pair representations. In other words, NSP is a classification head on top of the BERT model. We measure cohesiveness or if the rewrite follows the context via the probabilities obtained from the NSP head of the BERT model. We refer the reviewer to the BERT paper [7] and huggingface documentation [8] for additional details.
> \
> \
> [7] Devlin, Jacob, Ming-Wei Chang, Kenton Lee, and Kristina Toutanova. "Bert: Pre-training of deep bidirectional transformers for language understanding." arXiv preprint arXiv:1810.04805 (2018).\
> [8] https://huggingface.co/docs/transformers/model_doc/bert#transformers.BertForNextSentencePrediction
> \
> &nbsp;
> 4. `Sensitivity analysis for hyperparameter alpha in CtxSimFit`\
> We chose to present CtxSimFit with alpha=0.5 in the paper to emulate an equal weightage of contextual cohesiveness and semantic similarity to the original sentence (meaning preservation) for a general case of a stylistic rewrite. We performed a preliminary sensitivity analysis of the hyperparameter alpha: \
> CtxSimFit = $\alpha$∗BERTSCORE(S,X) + (1-$\alpha$) * NSP(C, X)
> \
> A) When alpha is small ($\alpha$ < 0.5), contextual cohesiveness measured by BERT next sentence prediction (NSP) is emphasized more. Contextual rewrites are scored higher compared to non-contextual rewrites. The Spearman correlation with human judgments of 'overall fit' ranges from 0.65-0.77, indicating alignment with human preferences.
> \
> B) When $\alpha$=0.5 (presented in the paper), there is an equal weightage on the contextual cohesiveness (NSP) and semantic similarity via BERTSCORE. Spearman correlation with human judgments of ‘overall fit’ is $\rho$ = 0.85 $p < 0.01$, indicating even stronger alignment with humans than smaller alpha values.
> \
> C) When alpha is large ($\alpha$ > 0.5), there is a stronger focus on preserving the meaning of the original sentence (non-contextual metric), resulting in comparable scores for both contextual and non-contextual rewrites. The Spearman correlation $\rho$ with human judgments of 'overall fit' is slightly significant ($p < 0.05$) for alpha values close to 0.5 but not significant for values farther away, indicating a lack of alignment with human preferences.
> \
> We refer the reviewer to response 1 to Reviewer [XbBa](https://openreview.net/forum?id=In4L79U5n7&noteId=43bvMBdDLS) for additional details.
> \
> &nbsp;
> 5. `Intuition for CtxSimFit`\
> We frame CtxSimFit as the linearly weighted combination of contextual cohesiveness and semantic similarity. The combination of these two elements through a linear projection allows for improved human alignment within both vector representation spaces. Furthermore, the hyperparameter alpha offers users better control over their preferences for contextual cohesiveness versus semantic similarity.
> \
> We refer the reviewer to response 2 to Reviewer [XbBa](https://openreview.net/forum?id=In4L79U5n7&noteId=43bvMBdDLS) for additional details.
> \
> &nbsp;
> 6. `Reasoning for lower correlation of CtxSimFit score in the Sentiment task`\
> Section 5.2 of our paper discusses and supports the notion, as indicated by prior research [1], that the Sentiment style transfer task may be ill-defined. Though the task of stylistic text rewriting is to maintain the original intended meaning while adapting to a different style, sentiment style transfer demands a change in both meaning and style (esp. in reviews where the reason why a review was written was to convey the original sentiment).  We believe this is the reason why the results of human evaluation for the sentiment rewriting task have lower annotator agreement compared to toxicity and formality tasks. Furthermore, the low agreement scores could explain the generally low correlation of our CtxSimFit metric with human preferences.
> \
> \
> For future work, we believe there is a need for a separate set of evaluation metrics that attempt to extract event/entity-specific content, and event-independent content and weigh these signals according to the task requirements.  Some preliminary work has been done in this area [9], and we hope that our work motivates further research in this field.
> \
> \
> [9] Yu, Ping, Yang Zhao, Chunyuan Li, and Changyou Chen. "Rethinking sentiment style transfer." In Findings of the Association for Computational Linguistics: EMNLP 2021, pp. 1569-1582. 2021.
> \
> &nbsp;
> 7. `Taxonomy of errors made by (non)contextual models`\
> We briefly discuss this in our paper under Section 8 - Context is crucial for rewriting.  Non-contextual models often struggle with certain aspects that contextual rewriting models excel at.
> \
> For formality style transfer, non-contextual models may have difficulty retaining context details, disambiguating entities (as shown in example (a) in Table 2), enhancing cohesiveness (as demonstrated in example (b)), and improving vocabulary usage (as seen in example (c)).
> \
> In detoxification, non-contextual models may fail to preserve intended meanings (as illustrated in example (d)), maintain politeness consistency (as showcased in example (e)), and show demographic sensitivity by generating less offensive rewrites (shown in example (f)).
> \
> Similarly, in sentiment transfer, non-contextual models may struggle to resolve ambiguity caused by sentiment change (as exemplified in example (g)) and fail to generate engaging rewrites (as seen in examples (h) and (i)).
> We will elaborate on this qualitative analysis in the final version of the paper.
> \
> &nbsp;
> 8. `Presentation improvements`\
> Thank you for your feedback, we will incorporate these tables in the main text.

---

### Official Review · Reviewer_joB3 · 2023-08-05

**Soundness:** 4

**Excitement:**

4: Strong: This paper deepens the understanding of some phenomenon or lowers the barriers to an existing research direction.

**Paper Topic And Main Contributions:**

This paper brings attention to how most existing work on text revision focuses on the text to be revised and not its surrounding sentences. It talks about how context is an important factor for humans, both when rewriting text or evaluating revisions, and suggests a new metric that takes original sentence similarity and contextual coherence into account. The authors apply their work to multiple datasets that provide different styles with respect to formality, toxicity, and sentiment. The revised sentences are collected from two GPT models and preference data of whether the annotator prefers contextual or non-contextual edits are collected from MTurk. Their findings show that humans prefer contextual rewrites and the metric proposed here aligns more strongly with human preference.

**Questions For The Authors:**

A. Line 326: Does this n-gram comparison include the context? If not, I wonder if it really makes sense to penalize contextual rewrites that borrowed words not in the original sentence from the context.

B. Would human preference drastically change if you removed sentence context? I'm wondering whether the consistency in meaning strongly overshadows other characteristics that make up a good sentence. (For examples, are humans more forgiving with grammatical errors with the contextual rewrites?)

**Reasons To Accept:**

I agree that current, most commonly used automatic text revision metrics are lacking, and would be interested in applying their new metric. Collecting human preference data can be time-consuming and expensive and having a metric that is easier to use but is also correlated with crowdsourced data would be useful to the community. The paper is also well-written and easy to read.

**Reasons To Reject:**

While not perfect, automatic text revision metrics are an established method that have been used widely for evaluating text. Claiming that it's not correlated with human preference is a strong statement given the size of the test set and there is no evidence to show that this claim holds true outside of evaluating formality, toxicity, and sentiment.



**Reproducibility:**

4: Could mostly reproduce the results, but there may be some variation because of sample variance or minor variations in their interpretation of the protocol or method.

**Reviewer Confidence:**

3: Pretty sure, but there's a chance I missed something. Although I have a good feel for this area in general, I did not carefully check the paper's details, e.g., the math, experimental design, or novelty.

**Typos Grammar Style And Presentation Improvements:**

A. Figure 2: It would be helpful to briefly mention that the numbers in the bars are representing the proportion of those who voted for a particular category.

---

> ### Author Rebuttal · Authors · 2023-08-29
>
> We thank the reviewer for the thoughtful comments and positive feedback on “having a metric that is easier to use but is also correlated with crowdsourced data would be useful to the community”.
>
> 1. `Existing non-contextual automatic metric correlation with human preferences`\
> \
> We agree with the reviewer that these automatic metrics are widely used in evaluating stylistic rewriting. However, what we illustrated in our paper is that these widely used metrics have focused on sentence/statement level non-contextual evaluation, often failing to align to account for the preceding textual context which is necessary for better representing realistic downstream use cases [1].
> \
> \
> We illustrate this via our contextual human evaluation study. Though we agree with the reviewer that the size of the test set is small, we show that **our results are statistically significant** through a binomial test. With these statistically significant human preference results, we compare correlations (Spearman and Kendall tau) of both non-contextual and our proposed contextualized automatic metrics with human preferences of ‘overall fit’ and ‘naturalness’ dimensions, across three stylistic rewriting tasks.
> \
> \
> We agree with the reviewer that we demonstrate the shortcomings of existing non-contextual automatic only for the three tasks - formality, sentiment, and detoxification. We will modify the language in the paper to reflect and clarify this. However, we are optimistic about the relevance and need for contextual automatic metrics in other generation tasks, given initial work on the relevance of context in evaluating open-domain dialogue generation [2, 3].
> \
> \
> [1] Yeh, Yi-Ting, Maxine Eskenazi, and Shikib Mehri. "A Comprehensive Assessment of Dialog Evaluation Metrics." In The First Workshop on Evaluations and Assessments of Neural Conversation Systems, pp. 15-33. 2021.\
> [2] Welleck, Sean, Jason Weston, Arthur Szlam, and Kyunghyun Cho. "Dialogue Natural Language Inference." In Proceedings of the 57th Annual Meeting of the Association for Computational Linguistics, pp. 3731-3741. 2019.\
> [3] Pang, Bo, Erik Nijkamp, Wenjuan Han, Linqi Zhou, Yixian Liu, and Kewei Tu. "Towards Holistic and Automatic Evaluation of Open-Domain Dialogue Generation." In Proceedings of the 58th Annual Meeting of the Association for Computational Linguistics, pp. 3619-3629. 2020.\
> \
> &nbsp;
> 2. `L326 Event-level Similarity: “Does this n-gram comparison include the context?” `\
> \
> We define “Event-level Similarity” ranking as “the rewrite that most effectively retains the essential events, entities, and relations present in the original sentence” (L272), and thus **does not include the context**. We intentionally do not include context for “event-level similarity” since we believe that non-contextual lexical and semantic automatic metrics emulate this dimension. We hoped to separate and contrast it against the “overall fit” and “naturalness” dimensions, which do consider context, to see the effect of the inclusion of extra entity/event details in contextual rewrites on the overall preference of human annotators. Through this experiment, we showed that non-contextual rewrites are most aligned with the entity/event details in the original sentence, yet annotators still prefer contextual rewrites for their naturalness and fit. We agree with the reviewer that contextual rewrites should not be penalized for “words not in the original sentence from the context” - hence our argument for inclusion and usage of contextual automatic metrics for stylistic rewriting as existing non-contextual metrics do indeed penalize contextual rewrites for including details from the context (Section 6).
> \
> &nbsp;
> 3. `Would human preference drastically change if you removed sentence context? Does “consistency in meaning strongly overshadow other characteristics that make up a good sentence” `\
> Yes, we believe that human preferences would change if we excluded the previous context and the results would be similar to the “event-level similarity” preference dimension where non-contextual rewrites are preferred to contextual rewrites. We agree with the reviewer that there is a greater emphasis on “meaning” in our human evaluation experiment, and lesser emphasis on other characteristics like grammatical errors (which are very uncommon with modern LLMs). However, we consider “Fluency” via non-contextual perplexity (Section 6.1 - Metrics considered for non-contextual automatic evaluation) as a proxy to measure grammatical errors and sentence quality [4, 5, 6]. We see that both contextual and non-contextual rewrites generate decently fluent rewrites, regardless of context.
> \
> \
> [4] Holtzman, Ari, Jan Buys, Maxwell Forbes, Antoine Bosselut, David Golub, and Yejin Choi. "Learning to Write with Cooperative Discriminators." In Proceedings of the 56th Annual Meeting of the Association for Computational Linguistics (Volume 1: Long Papers), pp. 1638-1649. 2018.\
> [5] Xu, Jingjing, Xuancheng Ren, Junyang Lin, and Xu Sun. "Diversity-promoting GAN: A cross-entropy based generative adversarial network for diversified text generation." In Proceedings of the 2018 conference on empirical methods in natural language processing, pp. 3940-3949. 2018.\
> [6] Ma, Xinyao, Maarten Sap, Hannah Rashkin, and Yejin Choi. "PowerTransformer: Unsupervised Controllable Revision for Biased Language Correction." In Proceedings of the 2020 Conference on Empirical Methods in Natural Language Processing (EMNLP), pp. 7426-7441. 2020.\
> \
> &nbsp;
> 4. `Presentation improvements`\
> Thank you for your feedback, we will address the figure beautification change.

---

### Official Review · Reviewer_XbBa · 2023-08-07

**Soundness:** 4

**Excitement:**

4: Strong: This paper deepens the understanding of some phenomenon or lowers the barriers to an existing research direction.

**Paper Topic And Main Contributions:**

This paper claims the empirical need to incorporate context information in stylistic text rewriting tasks, not only in the text generation but also (and more importantly) in the evaluation stage. To support this argument, three rewriting tasks (formality transfer, sentiment transfer, toxicity adjustment) over two types of contexts (conversation and document setting) are explored, and three human evaluators validated the human-level perception of the rewrites. Also, this paper finds out that the correlation between human perception and pre-existing metrics can be improved with a newly proposed evaluation metric ``CtxSimFit''.


**Reasons To Accept:**

This paper contains a thoroughly evaluated empirical claims, with a wide coverage of tasks and perspectives. The comparisons of different metrics and their correlations to human perceptions are presented in an easy-to-read manner, with clear writing. Despite the simple and straightforward method to incorporate textual context (and lack of alternative methods to compare), the method itself is evaluated thoroughly in design and also solidly enough in statistical manners to convince the readers the need of contextual element in stylistic rewrite tasks. Lastly, the presentation of what can be supported in the main text and the clear listing of limitations make the reader easy to determine the exact coverage of the research presented in this paper.


**Reasons To Reject:**

As CtxSimFit is a newly proposed metric that combines BERTSCORE and probability from BERT NSP head, a careful empirical validation on how sensitive the metric is to the relative weight hyperparameter alpha should have been presented. Also, Albeit a minor issue, the overall depth of insight would have been deeper, had there been theoretical exploration on the underlying meaning or interpretation of the linear combination of the two elements consisting the new metric CtxSimFit.


**Reproducibility:**

3: Could reproduce the results with some difficulty. The settings of parameters are underspecified or subjectively determined; the training/evaluation data are not widely available.

**Reviewer Confidence:**

2: Willing to defend my evaluation, but it is fairly likely that I missed some details, didn't understand some central points, or can't be sure about the novelty of the work.

---

> ### Author Rebuttal · Authors · 2023-08-29
>
> We thank the reviewer’s insightful comments and for the overall positive endorsement of our paper.
>
> 1. `Sensitivity and intuition for the hyperparameter alpha in CtxSimFit`\
> \
> Thank you for this question. We chose to present CtxSimFit with alpha=0.5 in the paper to emulate an equal weightage of contextual cohesiveness and semantic similarity to the original sentence (meaning preservation) for a general case of a stylistic rewrite. However, we agree that different use cases might require different weightings of both components, which we will add to the paper. We present the reviewer with some preliminary sensitivity analysis of the hyperparameter alpha: CtxSimFit = $\alpha$∗BERTSCORE(S,X) + (1-$\alpha$) * NSP(C, X)
> \
> \
> A) When alpha is small ($\alpha$ < 0.5): there is more emphasis on the contextual cohesiveness measured by the BERT next sentence prediction (NSP). We notice that contextual rewrites are scored much higher than non-contextual rewrites. Further, the Spearman correlation  $\rho$ with human judgments of ‘overall fit’ is significant and around the range of 0.65-0.77, suggesting alignment with human preferences.
> \
> For example for $\alpha=0.2$, the Spearman $\rho$ = 0.74, $p < 0.01$ (averaged across all tasks).
> \
> \
> B) When $\alpha$=0.5 (presented in the paper): there is an equal weightage on the contextual cohesiveness (NSP) and semantic similarity via BERTSCORE. Spearman correlation with human judgments of ‘overall fit’ is $\rho$ = 0.85 $p < 0.01$, showing stronger alignment with humans than smaller alpha.
> \
> \
> C) When alpha is large ($\alpha$ > 0.5):  there is a greater emphasis on the semantic similarity to the original sentence i.e. meaning preservation (non-contextual metric), hence the contextual and non-contextual rewrites are scored fairly equally by our metric. Further, the Spearman correlation  $\rho$ with human judgments of ‘overall fit’ is slightly significant  $p < 0.05$ for alpha values closer to 0.5 and non-significant farther away from 0.5, suggesting that this isn't well-aligned with humans' preferences.
> \
> For example for $\alpha=0.7$, the Spearman $\rho$ = 0.46, $p < 0.05$ (averaged across all tasks).
> \
> \
> We will expand upon this and include details about hyperparameter sensitivity in the final version of the paper.
> \
> &nbsp;
> 2. `Underlying meaning or interpretation of the linear combination of the two elements in CtxSimFit` \
> \
> We propose CtxSimFit as a metric that combines contextual cohesiveness and semantic similarity to assess the quality of a rewritten sentence. The linear combination of these two elements offers a projection into both vector representation spaces and, therefore could lead to better human alignment. This approach is commonly used in NLP, as seen in the BERT model where token, position, and segment embeddings are added to obtain a richer input representation. In our case, we are considering a one-dimensional scenario.
> \
> \
> We include the hyperparameter alpha to allow users to have better control over their preferences for meaning preservation vs. contextual fit. This is particularly useful in creative use cases of stylistic rewriting, where users may prioritize contextual cohesiveness over content preservation. On the other hand, for tasks such as formality transfer, users may value meaning preservation more than they would for detoxification tasks.
> \
> &nbsp;
> 3. `Clarification on the lack of alternative methods to compare`\
>  \
> We acknowledge that we relied on few-shot prompting of LLMs to incorporate textual context. We specifically chose LLMs because they have demonstrated strong rewriting capabilities, both with and without the use of textual context  [1]. Other existing generative models, such as those used for chit-chat and goal-oriented conversational agents, as well as pretrained language models, have struggled with effectively understanding and utilizing preceding textual context [2, 3, 4, 5]. Conversely, custom-made rewriting models from prior research often lack the modeling of context [6, 7]. Therefore, our decision to focus on LLMs was made as an initial investigation. We will add the lack of alternate custom or smaller models in our limitations section.
>
> _References:_
>
> [1] Brown, Tom, Benjamin Mann, Nick Ryder, Melanie Subbiah, Jared D. Kaplan, Prafulla Dhariwal, Arvind Neelakantan et al. "Language models are few-shot learners." Advances in neural information processing systems 33 (2020): 1877-1901.
>
> [2] Sankar, Chinnadhurai, Sandeep Subramanian, Christopher Pal, Sarath Chandar, and Yoshua Bengio. "Do Neural Dialog Systems Use the Conversation History Effectively? An Empirical Study." In Proceedings of the 57th Annual Meeting of the Association for Computational Linguistics, pp. 32-37. 2019.
>
> [3] O’Connor, Joe, and Jacob Andreas. "What Context Features Can Transformer Language Models Use?." In Proceedings of the 59th Annual Meeting of the Association for Computational Linguistics and the 11th International Joint Conference on Natural Language Processing (Volume 1: Long Papers), pp. 851-864. 2021.
>
> [4] Parthasarathi, Prasanna, Joelle Pineau, and Sarath Chandar. "Do Encoder Representations of Generative Dialogue Models have sufficient summary of the Information about the task?." In Proceedings of the 22nd Annual Meeting of the Special Interest Group on Discourse and Dialogue, pp. 477-488. 2021.
>
> [5] Su, Hsuan, Shachi H. Kumar, Sahisnu Mazumder, Wenda Chen, Ramesh Manuvinakurike, Eda Okur, Saurav Sahay, Lama Nachman, Shang-Tse Chen, and Hung-yi Lee. "Position Matters! Empirical Study of Order Effect in Knowledge-grounded Dialogue." arXiv preprint arXiv:2302.05888 (2023).
>
> [6] Ma, Xinyao, Maarten Sap, Hannah Rashkin, and Yejin Choi. "PowerTransformer: Unsupervised Controllable Revision for Biased Language Correction." In Proceedings of the 2020 Conference on Empirical Methods in Natural Language Processing (EMNLP), pp. 7426-7441. 2020.
>
> [7] Dale, David, Anton Voronov, Daryna Dementieva, Varvara Logacheva, Olga Kozlova, Nikita Semenov, and Alexander Panchenko. "Text Detoxification using Large Pre-trained Neural Models." In Proceedings of the 2021 Conference on Empirical Methods in Natural Language Processing, pp. 7979-7996. 2021.

---

### Official Review · Reviewer_5HXe · 2023-08-10

**Soundness:** 4

**Excitement:**

4: Strong: This paper deepens the understanding of some phenomenon or lowers the barriers to an existing research direction.

**Paper Topic And Main Contributions:**

This paper proposes to integrate textual context into rewriting and evaluation stages of stylistic text rewriting, and propose a new evaluation metric: CtxSimFit.
They show that human prefers contextual rewrites over non-contextual ones, and the contextual evaluation metric correlates better to human evaluation.

**Reasons To Accept:**

1. The paper defined and proposed a new question, that is contextual style transfer and find it is better aligned to human preference compared to previous non-contextual approaches.
2. The paper investigated the comparison with non-contextual evaluation metrics, and found they were not correlated well to the human annotations, which is none of surprise though, and they propose a way to integrate context to improve these metrics.
3. The paper also proposes a new metric that both considers the semantic similarity and cohesiveness, that yield the best results. It would be useful for evaluation community.
4. The paper is well organised and written. Nice and clear tables and figures.
5. The experiment is sound, three tasks are considered, and a lot experiments are done, also the analysis is good to me.

**Reasons To Reject:**

No reason to reject in my opinion.

**Reproducibility:**

4: Could mostly reproduce the results, but there may be some variation because of sample variance or minor variations in their interpretation of the protocol or method.

**Reviewer Confidence:**

4: Quite sure. I tried to check the important points carefully. It's unlikely, though conceivable, that I missed something that should affect my ratings.

---

> ### Author Rebuttal · Authors · 2023-08-29
>
> We appreciate the reviewer's positive feedback on our paper, especially for considering that our novel metric  "would be useful for evaluation community".

---

### Meta-Review · Area_Chair_32tQ · 2023-09-26

**Recommendation:** 4

**Metareview:**

This paper highlights the importance of context in stylistic text rewriting, proposing the integration of textual context into rewriting and evaluation stages. It introduces the CtxSimFit evaluation metric, showing that human preference aligns with contextual rewrites and that this metric better correlates with human evaluation. The research explores various rewriting tasks and contexts, emphasizing the significance of contextual modeling in stylistic rewriting.

Reasons to accept:
- The paper introduces the concept of contextual style transfer, which is better aligned with human preference compared to previous non-contextual approaches, bringing a novel perspective to the field.
- Thorough investigation of non-contextual evaluation metrics and their lack of correlation with human annotations, along with a proposal to integrate context for improvement.
-Introduction of a new metric, CtxSimFit, which considers both semantic similarity and cohesiveness and demonstrates superior results, offering a valuable addition to the evaluation community.
-The paper is well-organized and well-written, featuring clear tables and figures that enhance readability.
-Sound experimentation with three tasks, extensive experiments, and insightful analysis.
-Clear presentation of research scope and limitations, aiding readers in understanding the paper's coverage and contributions.

Reasons to reject:
-Lack of empirical validation regarding the sensitivity of CtxSimFit metric to the relative weight hyperparameter alpha, and a need for theoretical exploration of the metric's underlying meaning.
-Insufficient evidence to support the claim that existing automatic text revision metrics are not correlated with human preference beyond the specific evaluation of formality, toxicity, and sentiment.
-Potential issues with the use of large language models (LLMs) and in-context learning without a clear explanation of their selection and their applicability to smaller models.
-Unclear definitions of formal linguistic concepts such as "Coherence" and "Cohesiveness,".
-Lack of clarification and insight regarding the NSP head and the choice of alpha=0.5 in the formula.

---

### Decision · Program_Chairs · 2023-10-07

**Decision:**

Accept-Main

**Comment:**

This paper highlights the importance of context in stylistic text rewriting, proposing the integration of textual context into rewriting and evaluation stages. It introduces the CtxSimFit evaluation metric, showing that human preference aligns with contextual rewrites and that this metric better correlates with human evaluation. The research explores various rewriting tasks and contexts, emphasizing the significance of contextual modeling in stylistic rewriting.

Reasons to accept:
- The paper introduces the concept of contextual style transfer, which is better aligned with human preference compared to previous non-contextual approaches, bringing a novel perspective to the field.
- Thorough investigation of non-contextual evaluation metrics and their lack of correlation with human annotations, along with a proposal to integrate context for improvement.
-Introduction of a new metric, CtxSimFit, which considers both semantic similarity and cohesiveness and demonstrates superior results, offering a valuable addition to the evaluation community.
-The paper is well-organized and well-written, featuring clear tables and figures that enhance readability.
-Sound experimentation with three tasks, extensive experiments, and insightful analysis.
-Clear presentation of research scope and limitations, aiding readers in understanding the paper's coverage and contributions.

Reasons to reject:
-Lack of empirical validation regarding the sensitivity of CtxSimFit metric to the relative weight hyperparameter alpha, and a need for theoretical exploration of the metric's underlying meaning.
-Insufficient evidence to support the claim that existing automatic text revision metrics are not correlated with human preference beyond the specific evaluation of formality, toxicity, and sentiment.
-Potential issues with the use of large language models (LLMs) and in-context learning without a clear explanation of their selection and their applicability to smaller models.
-Unclear definitions of formal linguistic concepts such as "Coherence" and "Cohesiveness,".
-Lack of clarification and insight regarding the NSP head and the choice of alpha=0.5 in the formula.